# Efficient mate finding in planktonic copepods swimming in turbulence

**François-Gaël Michalec[1]\*, Itzhak Fouxon[1], Sami Souissi[2], Markus Holzner[3,4]**

[1]Institute of Environmental Engineering, ETH Zürich, Zürich, Switzerland; [2]Univ. Lille, CNRS, Univ. Littoral Côte d'Opale, UMR 8187 - LOG - Laboratoire d'Océanologie et de Géosciences, Station Marine de Wimereux, Université de Lille, Wimereux, France; [3]Swiss Federal Institute of Forest, Snow and Landscape Research, Birmensdorf, Switzerland; [4]Swiss Federal Institute of Aquatic Science and Technology, Dübendorf, Switzerland

**Abstract** Zooplankton live in dynamic environments where turbulence may challenge their limited swimming abilities. How this interferes with fundamental behavioral processes remains elusive. We reconstruct simultaneously the trajectories of flow tracers and calanoid copepods and we quantify their ability to find mates when ambient flow imposes physical constrains on their motion and impairs their olfactory orientation. We show that copepods achieve high encounter rates in turbulence due to the contribution of advection and vigorous swimming. Males further convert encounters within the perception radius to contacts and then to mating via directed motion toward nearby organisms within the short time frame of the encounter. Inertial effects do not result in preferential concentration, reducing the geometric collision kernel to the clearance rate, which we model accurately by superposing turbulent velocity and organism motion. This behavioral and physical coupling mechanism may account for the ability of copepods to reproduce in turbulent environments.

**\*For correspondence:**
michalec@ifu.baug.ethz.ch

**Competing interests:** The authors declare that no competing interests exist.

## Introduction

Zooplankton play pivotal roles in aquatic ecosystems. They channel nutrients and energy from the primary producers to higher trophic levels (*Cushing, 1975*), support the development of larger organisms including commercially important fishes (*Beaugrand et al., 2003*), and form an important component of the biological carbon pump (*Steinberg and Landry, 2017*). Many species of zooplankton are motile and propel themselves. Motility grants them the ability to navigate in the most favorable direction (*Genin et al., 2005*), to exploit their heterogeneous resource landscape (*Menden-Deuer and Grünbaum, 2006*), and to interact with other organisms, whether prey, mates, or predators (*Bagøien and Kiørboe, 2005*; *Kiørboe et al., 2009*; *Gemmell et al., 2012*). Motility in the zooplankton therefore mediates and governs processes that determine much of their individual fitness (*Kiørboe, 2008*; *Kiørboe, 2011*) and that influence the biological dynamics of the ecosystem at multiple scales. This has direct ramifications for global processes such as food web productivity and the cycling and export of carbon in the ocean.

The motion of zooplankton is shaped by the combination of their limited swimming abilities and the movement of the ambient fluid (*McManus and Woodson, 2012*). In marine and estuarine environments where turbulence is highly intermittent both spatially and temporally, plankton often experience large fluctuations in flow velocity that impose physical constraints on their locomotion (*Michalec et al., 2015a*) and that may therefore interfere with their ability to perform fundamental processes mediated by motility. Intuitively, it seems that an immediate ecological consequence of turbulence is a reduction in the individual and population fitness of plankton. Indeed, field studies show that zooplankton migrate downward to calmer layers when turbulence is strong at the surface

of the ocean (*Incze et al., 2001*), presumably to maintain perception capabilities and swimming strategies (*Visser et al., 2009*). However, despite its ecological significance, the nature of the interactions between turbulence and plankton motility in terms of fitness remains largely unknown. This limits our understanding of the links between turbulence levels in the environment and the distribution of plankton populations in the water column. In addition, while earlier views have usually considered flow and motility as decoupled processes in their contributions to individual fitness (*Visser et al., 2009*), recent observations indicate that plankton can actively modulate their motion in response to turbulence to improve their survival in flowing environments (*DiBacco et al., 2011*; *Michalec et al., 2017*; *Fuchs et al., 2018*). This reflects the need to better integrate the multifaceted coupling between physical forcing and self-motility in plankton ecology. However, up to date, the importance of these interactions is not well understood, primarily because of the considerable challenge of tracking organisms swimming and interacting in three dimensions and in turbulence (*Yen et al., 2008*). These technical difficulties have hindered the development of models and the testing of theoretical approaches against real empirical data. Consequently, there are currently no models that take into account explicitly the coupling between turbulence and behavior and that have been validated via experimental observations (*Rothschild and Osborn, 1988*; *Pécseli et al., 2010*).

In this study, we examine how turbulence affects the ability of calanoid copepods to find mates for reproduction. Calanoid copepods are the most abundant metazoans in the ocean and in estuaries, where they represent a pivotal component of the food web (*Beaugrand et al., 2003*). These small organisms reproduce sexually. This involves encounters between two organisms and therefore requires the quest for mates, an obviously challenging task in their dilute environment. Several behavioral strategies, known from studies conducted in calm water, improve mate finding in calanoid copepods. In species that use chemical communication for mating, females attract males from a distance by releasing pheromone trails that can persist for tens of seconds and extend several centimeters away from the female. Males detect and follow these trails, increasing their swimming velocity along the pheromone gradient until contact is made (*Katona, 1973*; *Doall et al., 1998*; *Yen et al., 1998*; *Bagøien and Kiørboe, 2005*). However, under conditions involving turbulence, pheromone trails are bound to elongate exponentially because of the incompressibility of the flow (*Batchelor, 1952*). They thin out rapidly, become too thin to be detectable, and break within seconds or less into disconnected filaments of concentrated cues separated by gaps with no detectable signal. Then, further mixing and molecular diffusion cause local variations of the odorant to vanish (*Shraiman and Siggia, 2000*). A similar reasoning applies for species that release clouds of pheromones, which start to erode at very low intensities of turbulence (*Kiørboe et al., 2005*). Sexual dimorphism in swimming patterns represents the second behavioral strategy. Studies conducted in still water show that females have more convoluted trajectories, while males are generally more directionally persistent (*Kiørboe and Bagøien, 2005*; *Michalec et al., 2015a*). The assumption is that this feature prevents resampling the same volume and therefore increases the probability for males to encounter females or the chemical signals they release (*Kiørboe and Bagøien, 2005*; *Visser and Kiørboe, 2006*). However, this strategy is not likely to significantly increase mating probability in turbulence, because ambient flow redistributes zooplankton in space irrespectively of their movement patterns in calm hydrodynamic conditions, and sexual dimorphism in motion patterns disappears even in weak turbulence (*Michalec et al., 2015a*). Consequently, the mechanisms by which these widespread and ecologically fundamental organisms find mates for sexual reproduction in flowing environments remain unknown.

Using an advanced particle tracking technique that allows observing organisms moving and interacting in three dimensions, we present experimental measurements of the relative velocity and spatial distribution of calanoid copepods swimming in turbulence. Inspired by the phenomenology of droplet coalescence in the atmosphere (*Falkovich et al., 2002*), we quantify the separate contributions of organism motility and physical effects due to turbulence to encounters between mates. We show that two key conditions for mating in turbulence are met. Firstly, copepods reach the radius of perception at higher rates than in still conditions due to the contribution of turbulence and active motion. Secondly, they detect and pursue nearby organisms entering their radius of perception within the short time frame of the encounter. The distance at which directed motion toward nearby mates takes place is similar in calm water and in turbulence, which reveals the surprising ability of copepods to differentiate between the hydrodynamic signals generated by a nearby conspecific and

the background noise generated by turbulence. This behavior is driven by males only and allows copepods to convert high encounter rates within the perception radius to frequent contacts between organisms, and therefore to maintain efficient mate finding even when turbulence is strong. We then show that contacts lead to actual mating. These results add to our understanding of plankton encounter rates in turbulence, where it is often assumed that after an initial increase, the number of successful encounters in turbulence decreases because of a decrease in perception distance and because of the inability of the organisms to react to each other within the short time frame of the encounter. This trend was first recognized in theoretical studies of fish larvae feeding on zooplankton (*MacKenzie et al., 1994*; *Kiørboe and MacKenzie, 1995*) and was later confirmed by laboratory and field data (*MacKenzie and Kiørboe, 2000*; *Pécseli et al., 2019*). Our results show that copepods are able to reach nearby individuals crossing their perception distance even in relatively strong turbulence, leading to contacts between mates and then to actual mating. We suggest that this behavioral mechanism provides copepods with evolutionary advantages in flowing environments. Building on our experimental data, we develop a semi-empirical model of plankton encounter rates that shows satisfactory quantitative agreement with our measurements. Our theoretical approach allows incorporating the influence of turbulent velocity differences, preferential concentration (due to the density, shape, and finite size of the organisms), and self-motility on the pairwise radial relative velocity and therefore on encounter rates down to the radius of perception. This study provides new insights on how zooplankton maintain fitness in dynamic environments. It illustrates how the coupling between plankton motility and the movement of the ambient fluid shapes fundamental biological processes at the base of the trophic network in aquatic ecosystems.

## Results and discussion

### Flow parameters

We measure simultaneously the motion of the calanoid copepod *Eurytemora affinis* and neutrally buoyant flow tracers in a device generating homogeneous isotropic turbulence via two panels of counter-rotating disks located on its lateral sides (*Figure 1A*) (see Materials and methods for additional details). We conduct measurements on adult males, adult females, and late-stage copepodites in approximately equal proportions, with a number density of one organism per cubic cm. Relevant flow parameters for the turbulence intensity are given in *Table 1*. The turbulent velocity achieved in our setup (5.6 mm s$^{-1}$) is moderate and comparable to values measured in the field (*Yamazaki and Squires, 1996*; *Pécseli et al., 2019*). The integral length scale (6 mm) and the Kolmogorov time and length scales (0.2 s and 0.45 mm) are well within the range considered in previous laboratory and theoretical studies of plankton motion in turbulence (*Yamazaki and Squires, 1996*; *Yen et al., 2008*) and observed in coastal environments (*Pécseli et al., 2019*). The measured energy dissipation $\epsilon = 3 \times 10^{-5}$ m$^2$ s$^{-3}$ is on the upper range of the intensities measured in the open ocean, where $\epsilon$ typically ranges from $10^{-8}$ m$^2$ s$^{-3}$ below the mixed layer to $10^{-4}$ m$^2$ s$^{-3}$ near the surface, and is comparable to values measured in environments inhabited by *E. affinis* such as coastal areas and estuaries where most of the turbulence originates from tidal forcing instead of wind (*Pécseli et al., 2019*). We note that selecting a turbulence intensity on the upper range of typical marine conditions sets an upper bound on the physical constraints placed by fluid flow on plankton motion, and is therefore appropriate to identify the behavioral strategies developed by these small organisms in their dynamic environments where turbulence is highly intermittent both spatially and temporally. In such environments, zooplankton can experience changes in $\epsilon$ varying over six orders of magnitude (*Yamazaki and Squires, 1996*).

### Behavior and physical processes drive plankton encounters in turbulence

We start our analysis by considering the mechanisms that govern the motion of plankton in turbulence and therefore drive their encounters. We define an encounter as any event where two individuals are separated by less than 1 mm (*Figure 1B*). This distance corresponds to approximately one body length of an adult *E. affinis* and is the minimal distance at which two closely interacting copepods with complex shape and motion could be tracked reliably. We stress that this minimal distance corresponds to the separation between the centroids of two adjacent copepods. This means that

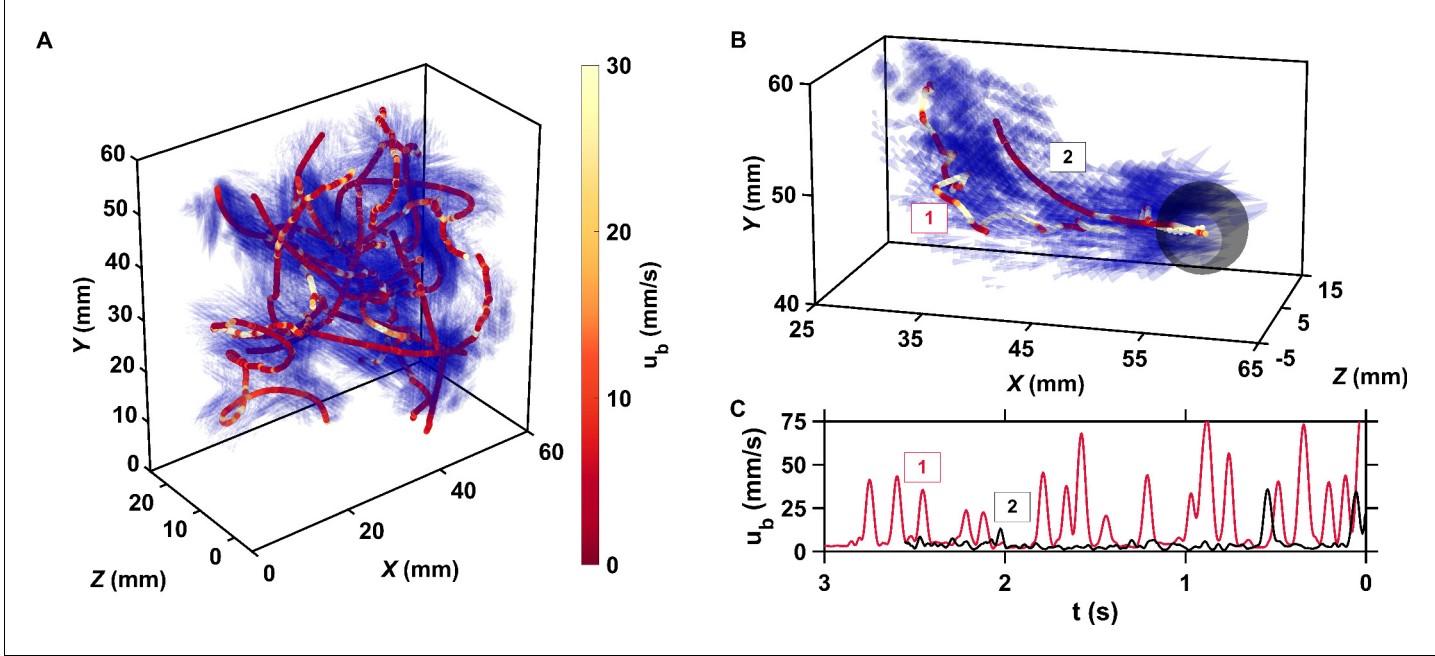

**Figure 1.** The motion of copepods in turbulence, and therefore their encounters, are governed by the coupling between active swimming and fluid motion. (A) Examples of trajectories of copepods swimming in turbulence, color-coded with the magnitude of their velocity $u_b$ with respect to the flow to emphasize the contribution of motility, and instantaneous flow field along the trajectories. The length of the cones is proportional to the flow velocity. Simultaneous measurements of tracer and copepod trajectories allow retrieving the behavioral component of their motion. Because the cones are semi-transparent to allow visualizing the copepod trajectories, it may be necessary to zoom in to see them individually. (B) Trajectories of two copepods up to the time of encounter, showing periods of vigorous swimming where $u_b$ is large, and periods of more passive motion dominated by turbulent advection where $u_b$ is negligible compared to the flow velocity. The gray sphere shows the perception volume, within which hydrodynamic perturbations from nearby organisms can be detected. As described below, organisms within the perception volume tend to move toward each other. (C) $u_b$ versus time to encounter $t$ for the two trajectories shown in panel B. Relocation jumps are clearly visible. They represent the active component of the motion and allow copepods to depart from the flow streamlines (*Michalec et al., 2017*).

two organisms whose centroids are separated by 1 mm are actually colliding and that we are able to observe encounters down to contact events. For every copepod involved in an encounter, we measure the distance $d_{drift}$ between its actual position along the observed trajectory and the position of a virtual particle transported passively by the flow. We solve for the position of this virtual particle backward in time by integrating the underlying flow field at each time instant, starting from the time of encounter and from the position of the copepod at the time of encounter. The distance $d_{drift}$ is by definition zero for a particle with vanishingly small diameter and inertia because its velocity is exactly that of the local instantaneous flow. In this case, the only mechanism governing particle motion and leading to encounter is advection by turbulence (*Appendix 1—figure 1*). Copepods, however, are comparable in size to the Kolmogorov length scale $\eta$, slightly heavier than seawater (*Knutsen et al., 2001*), and have a complex shape. The Stokes number $St$, defined as the ratio between particle response time and flow time scale, quantifies the importance of inertia. We use the definition of $St$ for large inertial particles, using the eddy turnover time at the scale of the particle as the relevant time scale of the flow (*Xu and Bodenschatz, 2008*). We find that $St \approx 0.1$ for inert carcasses at our

**Table 1.** Relevant turbulence parameters in the investigation volume.

$\epsilon$ is the space- and time-averaged turbulent energy dissipation rate. $\tau_\eta = (\nu/\epsilon)^{1/2}$ and $\eta = (\nu^3/\epsilon)^{1/4}$ are the Kolmogorov time and length scales, respectively. $u_{rms}$ is the root-mean-square of the velocity fluctuations. $R_\lambda$ is the Taylor-scale Reynolds number, and $L$ is the integral length scale.

| $\epsilon$ (m$^2$ s$^{-3}$) | $\tau_\eta$ (s) | $\eta$ (mm) | $u_{rms}$ (mm s$^{-1}$) | $R_\lambda$ | $L$ (mm) |
|---|---|---|---|---|---|
| $3 \times 10^{-5}$ | 0.2 | 0.45 | 5.6 | 22 | 6 |

intensity of turbulence. This value is larger than for tracers, and it is therefore not surprising that dead copepods depart more from the flow streamlines, as illustrated by the increasingly large $\langle d_{drif} \rangle$ of inert carcasses as the time to encounter increases (*Appendix 1—figure 1*). Living copepods depart even more (*Appendix 1—figure 1*) because in addition to inertial effects, they also self-propel: we have previously shown that copepods in turbulence perform frequent relocation jumps that enable them to reach velocities larger than the turbulent velocity in typical oceanic conditions (*Michalec et al., 2017*; *Figure 1C*). The dynamics of copepods swimming in turbulence is therefore governed by three mechanisms: advection by the flow and inertial effects due to the density, elongated shape, and finite size of the organisms, which are two physical processes, and naturally also their active motion. We investigate these mechanisms separately to quantify their relative contribution to encounters between mates.

## Self-locomotion in turbulence increases the flux of organisms into the perception volume

Turbulence enhances the rate at which particles encounter one another by increasing the inward component of their pairwise radial relative velocity $u_r = (v_2 - v_1) \cdot [(r_2 - r_1)/\|(r_2 - r_1)\|]$ where $v_1$ and $v_2$ are the velocity vectors of two particles at a given time, and $r_1$ and $r_2$ are their coordinate vectors. This mechanism is referred to as *turbulent velocity differences* and was originally defined for particles with no inertia, for which $u_r$ is entirely determined by the local flow velocity (*Saffman and Turner, 1956*). For $r = \|(r_2 - r_1)\|$ in the inertial subrange, the velocity difference equals the characteristic velocity of eddies of size $r$. At moderate Reynolds numbers where intermittency is not very strong, the velocity difference grows on average with $r$ following the relation $[\langle (v_2 - v_1)^2 \rangle]^{1/2} \sim r^{1/3}$ and is maximal for $r$ on the order of the integral length scale of the flow (*Frisch, 1995*). For living particles that self-propel, $u_r$ also includes their active motion.

To understand quantitatively how turbulent velocity differences and self-motility contribute to encounters, we compute $u_r$ for every unique pair of particles and at each time step along their trajectories. We average $u_r$ over all pairs that have an inward motion (i.e., $u_r < 0$) and condition it on the separation distance $r$ between the particles. The number of values used in the analysis ranges from 1,226,343 at $r = 1$ mm to 104,178,657 at $r = 20$ mm. We obtain the average inward pairwise radial relative velocity $\langle u_{r\,|\,i} \rangle$ that, assuming uniform concentration, gives the average clearance rate $\langle \gamma \rangle = 4\pi r^2 |\langle u_{r\,|\,i} \rangle|$, traditionally defined by marine biologists in the context of predator-prey interactions as the volume of water per time step that is visited by a predator. Multiplying $\gamma$ with the number density of prey gives the encounter rate, assuming that the predator is able to detect every prey that enters its clearance volume. It also gives the capture rate if every prey is captured with certainty. The radius $r$ of this volume, and therefore both the clearance and encounter rates, is determined by the perception capabilities of the predator. The same reasoning applies here with conspecific copepods in the context of mating.

We show $\langle \gamma \rangle$ as a function of $r$ in *Figure 2A* and estimate in *Figure 2B* the increase in encounter rate provided by active motion in turbulence as the ratio of the clearance rate of living copepods swimming in turbulence $\langle \gamma_{l,t} \rangle$ to that of dead copepods passively transported by the flow $\langle \gamma_{d,t} \rangle$. This ratio is larger than one at all $r$, indicating that the coupling between turbulent advection and active motion enhances encounter rates compared to advection alone. We attribute this increase to large differences in the inward component of the pairwise radial relative velocity of copepods, generated by their frequent relocation jumps in turbulence (*Michalec et al., 2017*). However, $\langle \gamma_{l,t} \rangle / \langle \gamma_{d,t} \rangle$ shows two distinct trends depending on the radius $r$ of the clearance volume. It increases monotonically as $r$ decreases down to approximately 4 mm, and rises sharply at shorter $r$ (*Figure 2B*). The transition occurs at a distance $r \approx 4$ mm that was previously reported to mark the onset of behavioral interactions mediated by hydrodynamic signals in calanoid copepods (*Bagøien and Kiørboe, 2005*). We refer to this distance as the *perception radius*, below which nearby conspecifics can be detected via the flow signals they generate. At $r$ corresponding to the perception radius, being motile in turbulence brings a significant (approximately twofold) increase in $\langle \gamma \rangle$ compared to advection alone (*Figure 2B*). We further show that $\langle \gamma_{l,t} \rangle$ for $r \geq 4$ mm is entirely determined by the superposition of turbulent advection and the motion of independent organisms. We compute a clearance rate $\langle \gamma \rangle_{flow+behavior}$ by superposing the pairwise radial relative velocity of the turbulent flow $u_{r,t}(r)$ at separation $r$, drawn randomly from its probability density function, to the pairwise radial relative velocity

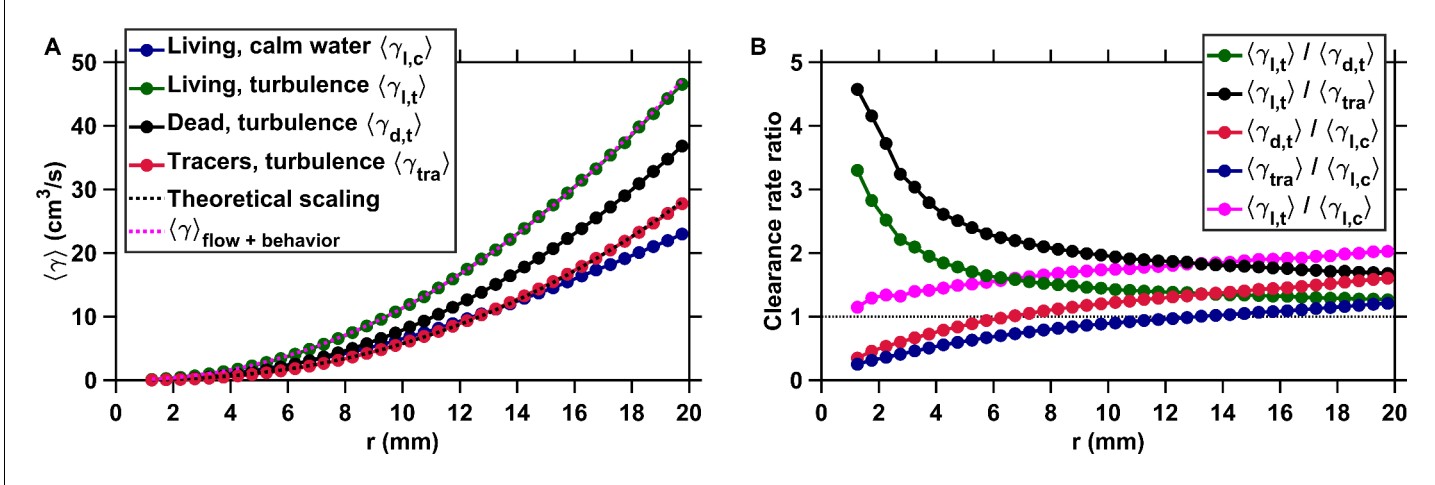

**Figure 2.** Active motion boosts encounter rates in turbulence. (**A**) Mean clearance rate $\langle\gamma\rangle$ versus the radius of the perception volume $r$ for tracers ($\langle\gamma_{tra}\rangle$, red), dead copepods passively transported by turbulence ($\langle\gamma_{d,t}\rangle$, black), and living copepods in turbulence ($\langle\gamma_{l,t}\rangle$, green) and in calm water ($\langle\gamma_{l,c}\rangle$, blue). $\langle\gamma\rangle$ is directly obtained from $\langle u_{r\,|\,i}\rangle$ measured for different values of the pairwise separation distance $r$. We verify the accuracy of our flow field measurements by plotting the theoretical prediction for the clearance rate of tracers, based on the $\epsilon^{1/3}r^{7/3}$ (for the inertial subrange) and $r^3(\epsilon/\nu)^{1/2}$ (for the viscous subrange) Kolmogorov scaling of the velocity differences (**Saffman and Turner, 1956**), using proportionality constants found empirically (**Pécseli et al., 2014**) (dashed line, black). The prediction agrees very well with our experimental data. We show that a very accurate estimation of $\langle\gamma_{l,t}\rangle$ down to the radius of hydrodynamic interactions can be achieved by considering the separate contributions of turbulent advection and the self-locomotion of two independent organisms. We compute $\langle\gamma\rangle_{flow+behavior}$ for living copepods in turbulence, using $u_r(r) = u_{r,t}(r) + u_{r,c,b}$ where $u_{r,t}(r)$ is randomly drawn from the probability density function of the pairwise radial relative velocity of tracers at a given separation distance $r$, and $u_{r,c,b}$ is the pairwise radial relative velocity of the copepods. $u_{r,c,b}$ is computed using randomly sampled values of the organism velocity with respect to the flow $u_b$ to isolate the behavioral part of their motion. $\langle\gamma\rangle_{flow+behavior}$ agrees very well with the measurements for $r \geq 4$ mm (dashed line, magenta). It deviates for shorter $r$ because of behavioral interactions within the radius of hydrodynamic interactions: the ratio $\langle\gamma\rangle_{flow+behavior}/\langle\gamma_{l,t}\rangle$ is approximately two at $r = 1$ mm and one at $r = 4$ mm. These interactions are studied in the next section. (**B**) Ratios of mean clearance rates. Active motion plays a preponderant role in enhancing encounter rates in turbulence, as evidenced by a ratio $\langle\gamma_{l,t}\rangle/\langle\gamma_{d,t}\rangle$ (green) larger than one, especially at short $r$ comparable to or below the radius of hydrodynamic interactions. $\langle\gamma_{l,t}\rangle$ results from turbulent velocity differences, organism motion, and inertia, while $\langle\gamma_{d,t}\rangle$ results from turbulent velocity differences and inertia only. The contribution of inertia to $\gamma$ is lower than that of self-locomotion but not negligible: the ratio $\langle\gamma_{l,t}\rangle/\langle\gamma_{tra}\rangle$ (black), where $\langle\gamma_{tra}\rangle$ is the mean clearance rate of small, neutrally buoyant flow tracers that have negligible inertia, is larger than $\langle\gamma_{l,t}\rangle/\langle\gamma_{d,t}\rangle$. This indicates that the combination of turbulent velocity differences and effects due to inertia leads to a larger $\gamma$ than turbulent velocity differences alone. We also note that while being passively transported by turbulence leads to a larger $\gamma$ at large $r$ compared to motility in still water, as evidenced by the ratios $\langle\gamma_{d,t}\rangle/\langle\gamma_{l,c}\rangle$ (red) and $\langle\gamma_{tra}\rangle/\langle\gamma_{l,c}\rangle$ (blue) above one for large $r$, it provides less mating opportunities than self-locomotion in calm hydrodynamic conditions at shorter $r$. This indicates that being passively transported by the flow is not an efficient mechanism to encounter many mates within the perception radius. It requires active swimming in turbulence for copepods to achieve an encounter rate that is comparable or even larger than in calm hydrodynamic conditions (ratio $\langle\gamma_{l,t}\rangle/\langle\gamma_{l,c}\rangle$, magenta).

The online version of this article includes the following source data for figure 2:

**Source data 1.** Source data for *Figure 2*.

of the copepods $u_{r,c,b}$, calculated using their velocity with respect to the flow and assuming independence between the velocity of the two organisms of the pair. *Figure 2A* shows that $\langle\gamma\rangle_{flow+behavior}$ agrees very well with $\langle\gamma_{l,t}\rangle$ for $r \geq 4$ mm, indicating that the process underpinning the increase in $\langle\gamma\rangle$ provided by motility in turbulence at $r$ above or equal to the perception radius is the independent self-locomotion of individual organisms.

## Inward motion within the clearance volume converts high encounter rates to actual contact events

The increase in $\langle\gamma\rangle$ provided by a larger $|\langle u_{r\,|\,i}\rangle|$ does not automatically translate into more mating events. As in predator-prey interactions, where the capture rate depends in part on the ability of the predator to catch prey entering its clearance volume (**Kiørboe et al., 2009**), a successful mating event also depends on the ability of a copepod to make contact with a conspecific within its perception volume. We show that a second mechanism, consisting in directed motion toward neighbor

organisms at short separations, allows copepods to convert the high encounter rates provided by motility in turbulence into actual contact events within the short time frame of the encounter. This behavior occurs within the perception radius for hydrodynamic signals (*Bagøien and Kiørboe, 2005*) and explains the sharp increase in the ratio $\langle \gamma_{l,t} \rangle / \langle \gamma_{tra} \rangle$ for $r \leq 4$ mm (*Figure 2B*).

We investigate this behavioral mechanism by plotting the radial relative velocity $u_r$ and the cosine of the approach angle $\cos(\theta) = [(\boldsymbol{v}_2 - \boldsymbol{v}_1)/\|(\boldsymbol{v}_2 - \boldsymbol{v}_1)\|] \cdot [(\boldsymbol{r}_2 - \boldsymbol{r}_1)/\|(\boldsymbol{r}_2 - \boldsymbol{r}_1)\|]$ conditioned on the pairwise separation distance $r$. $u_r$ includes both the radial inward and outward components and therefore it can be used as a proxy for the net average flux of particles experienced by another particle with perception distance r along its trajectory. $\cos(\theta)$ quantifies the alignment of the relative velocity vector with the pairwise separation vector. Negative values of $u_r$ and $\cos(\theta)$ indicate that the two particles of the pair move inward toward each other, positive values that they move outward away from each other. The probability density functions of $u_r$ and $\cos(\theta)$ are skewed toward negative values at short $r$ for living copepods in calm water and in turbulence, while they remain centered at zero for dead copepods and tracers in turbulence (*Figure 3*). Consequently, $\langle u_r \rangle$ and $\langle \cos(\theta) \rangle$ are both negative at short $r$ for living copepods. This gives clear evidence that, on average, two copepods move inward toward each other when they get in close proximity. We call this mechanism *collision efficiency*, drawing an analogy to the phenomenology of droplet collision in clouds, where hydrodynamic interactions between droplets become relevant at the last stage of the collision when their separation distance is comparable with their radii (*Falkovich et al., 2002*).

An interesting observation is that this behavior is driven by males only: similar measurements conducted in calm water with separate genders indicate clear short-range attraction in males but repulsion in females, since $\langle u_r \rangle_{\female} > 0$ at short separations (*Appendix 1—figure 2A*). Earlier studies

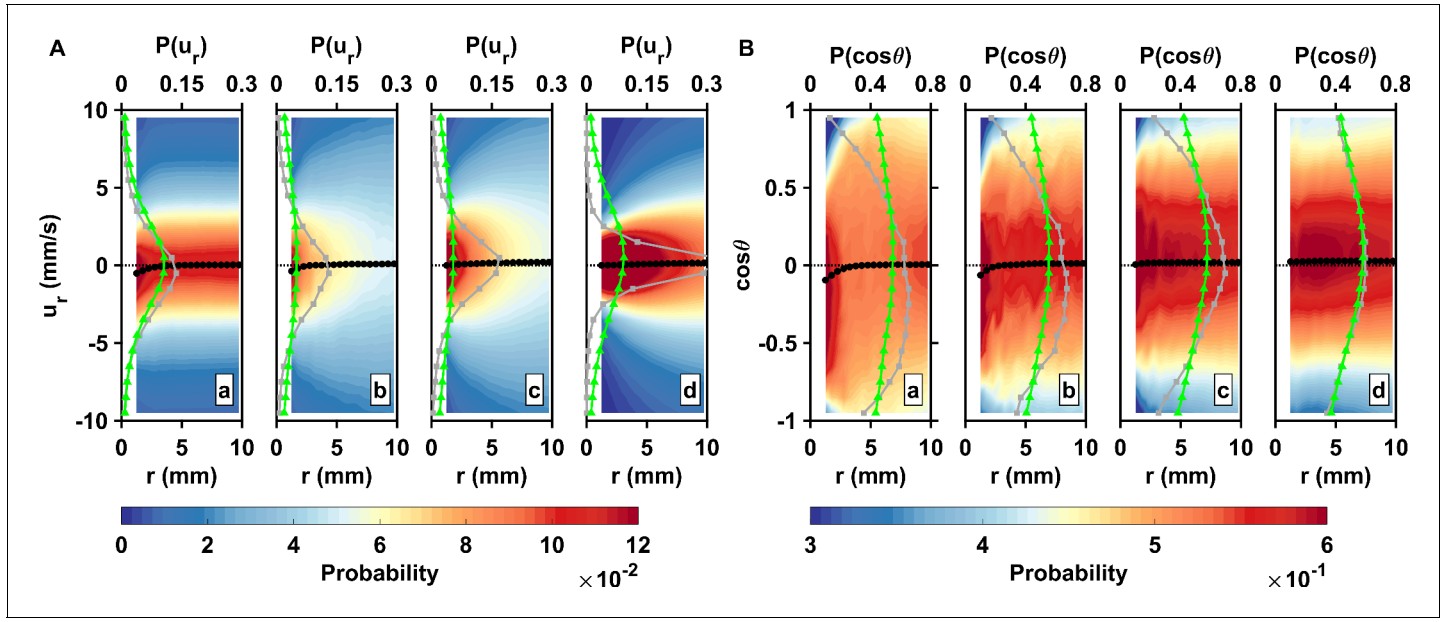

**Figure 3.** Inward motion within the perception distance converts high encounter rates brought by active motion into actual contact events. (**A**) Probability density functions of the pairwise radial relative velocity $u_r$ versus the separation distance $r$ for living copepods in calm water (a), living copepods in turbulence (b), inert carcasses in turbulence (c), and tracers in turbulence (d). The pairwise separation distance is along the horizontal axis, the velocity is along the vertical axis, and the probability is given by the color intensity. As an illustration for the shift in the distribution, the gray (squares) and green (triangles) curves show $P(u_r)$ at $r = 1$ mm and $r = 10$ mm, respectively. The black curve (circles) indicates the mean value at each bin of $r$. $u_r$ is negative when two particles move toward each other. $\langle u_r \rangle$ is negative at small $r$, indicating a net flux of organisms into the perception volume. (**B**) Probability density functions of the cosine of the approach angle $\theta$ versus the pairwise separation distance $r$, and mean values. $\theta$ is defined as the angle between the relative velocity vector and the particle separation vector. Negative values of $\cos(\theta)$ indicate that the two particles of the pair move inward toward each other. Note that the color scale has been truncated for better visibility. The gray and green curves show $P(\cos\theta)$ at $r = 1$ mm and $r = 10$ mm, respectively.

The online version of this article includes the following source data for figure 3:

**Source data 1.** Source data for *Figure 3*.

conducted in still water have shown that males lunge to catch females upon sensing the flow signals they generate, and that females jump away from their suitors (*Doall et al., 1998*; *Yen et al., 1998*; *Bagøien and Kiørboe, 2005*). This behavior constitutes the last step of the approach and occurs at a separation distance of a few body lengths (*Doall et al., 1998*; *Yen et al., 1998*; *Bagøien and Kiørboe, 2005*). The shift of $\langle u_r \rangle$ toward negative values for males and positive values for females observed in our measurements (*Appendix 1—figure 2A*) confirms these previous observations in still water and further indicates that males attempt to make contact with any conspecific within their perception radius, not only females, presumably to assess the gender of the neighbor organism based on contact glycoproteins at the surface of its cuticle (*Ting and Snell, 2003*). The net inward flux persists in turbulence when genders are present in equal proportions (*Figure 2A*) because $|\langle u_r \rangle|_{\male} > |\langle u_r \rangle|_{\female}$ at short $r$ (*Appendix 1—figure 2A*).

We confirm that the attraction between nearby organisms is entirely attributable to their self-locomotion by plotting $\langle u_r \rangle_{active}$, the pairwise radial relative velocity computed using the velocity of the copepods with respect to the underlying turbulent flow (i.e., the behavioral component of the motion), together with $\langle u_r \rangle_{passive}$, the pairwise radial relative velocity computed using the instantaneous flow velocity at the position of the organisms (i.e., the passive component of the motion) (*Appendix 1—figure 2B*). We show that $\langle u_r \rangle_{active}$ is negative at short $r$, whereas $\langle u_r \rangle_{passive}$ remains close to zero, as expected in the case of a purely behavioral process. Surprisingly, the distance $r \approx 4$ mm (corresponding to approximately 8 $\eta$) at which we observe directed motion between organisms is similar in calm water and in turbulence. This reveals the intriguing ability of copepods to correctly identify the hydrodynamic signals generated by a conspecific among the background noise caused by turbulence. We note that the actual correlation length $\eta_0$ of the velocity gradients in turbulence is usually several times larger than the dimensional estimate $\eta$ (*Ishihara et al., 2005*; *Pécseli et al., 2014*). This means that directed motion within the perception radius actually takes place within or over distances comparable to the viscous subrange where the velocity field is smooth. Our results show that, in this region, an organism appears to be able to distinguish between the flow disturbances created by another organism as it swims (*Kiørboe et al., 2014*) and the linear velocity gradients of turbulence within $\eta_0$. We also note that for turbulence intensities lower than that used in our measurements and often met in the water column under calm hydrodynamic conditions, all the encounter process takes place within the viscous subrange.

The inward motion of the male within the perception radius leads to contact between the two organisms of the pair. In some cases, this contact is brief and the male almost immediately disengages. We assume that this behavior occurs when the second organism is also a male. In other cases, the contact leads to mating. It was technically not feasible to resolve the motion of two individuals involved in mating via our automatic stereoscopic particle tracking technique. The reason is that once contact occurs, the two organisms often appear as overlapping pixel blobs in the images and consequently they are detected as a single particle during image segmentation. We however verified visually from the raw images that many mating events occurred and stereo-matched some of them by manually tracking the two organisms in the images. That is, we manually tracked the centroid of the two individuals in the images to avoid errors in the segmentation and plugged their pixels coordinates into the collinearity equations to obtain their three-dimensional coordinates via triangulation. We show in *Figure 4* a representative mating event obtained in this way, where a female swimming actively against the streamlines crosses the perception radius of a male that in this case is passively advected by the flow. It is visible that reorientation of the male occurs immediately after detection, then the male reaches the female by jumping while the female attempts at escaping. Earlier studies conducted in calm water have shown that, after successfully following a pheromone trail, male copepods make a final lunge for the female upon detecting the hydrodynamic signals it generates (*Doall et al., 1998*; *Yen et al., 1998*; *Bagøien and Kiørboe, 2005*). Here, we demonstrate that this fast lunging motion also occurs in turbulence within the very short time frame of the encounter. Active swimming and turbulence advection bring copepods within their perception distance, then males convert encounters to contacts via immediate reorientation and directed motion and then to actual mating. In general, we observed that during mating the pursuit is brief and contact always occurs within the perception radius. In some cases, the female struggles to break free from the male via vigorous jumps but the male successfully hangs on to the female. In other cases, as shown in *Figure 4*, the pair tumbles passively as it is advected by the turbulent flow. A large number of mating

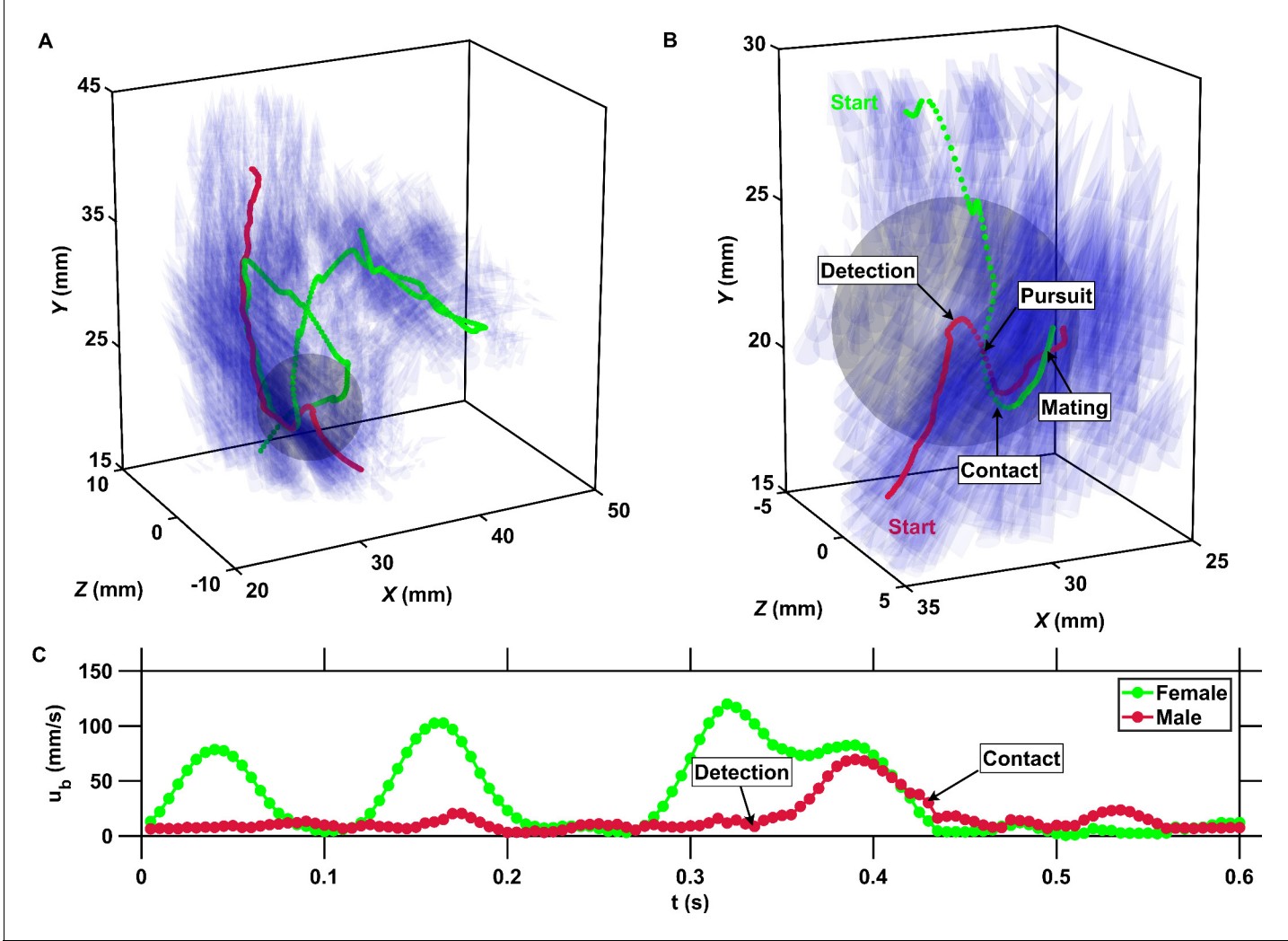

**Figure 4.** The inward motion of males within the perception radius leads to contact and mating. (A) Trajectory of a male (red) and a female (green) mating in turbulence, and instantaneous flow field (blue cones) along their trajectories. The length of the cones is proportional to the flow velocity. The cones are semi-transparent to allow visualizing the copepod trajectories. The gray sphere shows the perception volume of the male at the time of detection. (B) Close-up look at the encounter event for the same two trajectories. The female swims actively against the flow streamlines (blue cones oriented against the direction of motion) while the male is passively advected. The male detects the female crossing its perception volume. Reorientation and inward motion of the male occur immediately and lead to contact. During the pursuit, the male swims toward the female and against the flow streamlines. The male successfully reaches the female, after which the two organisms tumble and are advected passively by the flow. (C) Time series of the magnitude of the velocity $u_b$ with respect to the flow for the same two trajectories. In this example, the male is passively transported by turbulence while the female swims vigorously via relocation jumps. Upon detection, the male performs one single jump to reach the female, while the female unsuccessfully attempts at escaping.

events similar to the one shown in *Figure 4* are visible in the image sequences, indicating that mating does occur frequently in turbulence.

## Patchiness due to inertia does not contribute significantly to encounters between mates

We now examine the contribution of inertial effects due to density and size to understand further the mechanisms that drive mating encounters in turbulence. Effects due to inertia manifest themselves as the tendency of particles to distribute non-uniformly in turbulence. This phenomenon is referred to as *preferential concentration*. It can substantially increase the encounter rate of particles that have a different density from the carrier fluid and/or a finite size, because it causes particles that have comparable inertia to cluster in the same regions of the flow (*Reade and Collins, 2000*).

Preferential concentration is mainly driven by small-scale dissipative eddies. It has been intensively studied because it relates to important environmental and industrial processes, for instance the formation of rain droplets or the combustion of sprays (*Falkovich et al., 2002*). The focus has generally been on particles that are much smaller than $\eta$, the dissipative scale of the flow. Clustering is attributed to the centrifugation of small and heavy particles by turbulent vortices and their accumulation within regions of high strain rate outside the vortices (*Sundaram and Collins, 1997*). Large and heavy particles also tend to cluster in strain-dominated regions of the flow (*Guala et al., 2008*). On the opposite, small and light particles preferentially concentrate in regions of high vorticity (*Fouxon, 2012*; *Tagawa et al., 2012*). Much less is known about particles such as copepods, with sizes larger than or comparable to $\eta$ and densities close to that of the fluid. Previous theoretical work suggests that clustering due to inertia may be ecologically significant for zooplankton because it can increase the probability for two organisms to be within their perception radius, thereby facilitating mating (*Schmitt and Seuront, 2008*).

We quantify preferential concentration in copepods by considering the correlation of the particle concentration $f(\boldsymbol{x}, \boldsymbol{r}) = \langle n(\boldsymbol{x})n(\boldsymbol{x}+\boldsymbol{r})\rangle / [\langle n(\boldsymbol{x})\rangle\langle n(\boldsymbol{x}+\boldsymbol{r})\rangle]$ where $n(\boldsymbol{x})$ is the particle concentration within a 1 mm size cubic voxel centered at the position $\boldsymbol{x}$ and $\boldsymbol{r}$ is a separation vector. The voxel size corresponds to the average length of an adult *E. affinis* and is therefore appropriate to detect clustering at the scale of the organism. We note that this quantity is statistically similar to the pair correlation function $g(r)$ that accounts for clustering effects in the formulation of the geometric collision kernel $K(r) = \gamma(r)g(r)$. However, $f(\boldsymbol{x}, \boldsymbol{r})$ is less challenging to resolve accurately than $g(r)$ when the suspension is dilute, because $g(r)$ is very slow to converge. The correlation of the particle concentration is by definition one for uniformly distributed particles and higher than one for particles that cluster. *Figure 5* shows that $f(\boldsymbol{x}, \boldsymbol{r})$ remains very close to one at all $r$ for tracers, which is expected because these particles are not supposed to cluster: they are smaller than $\eta$ and have negligible inertia. For inert carcasses, $f(\boldsymbol{x}, \boldsymbol{r})$ increases very slightly at small $r$ (below approximately 2 mm) because of interactions between eddies in the flow at small scales and particle density, size, and elongated shape (*Figure 5*). However, the increase is negligible when compared to that observed in earlier studies with small ($d_p \ll \eta$) and heavy or light particles, for which $g(r)$ may increase by an order of magnitude at moderate intensities of turbulence similar to that used in our measurements (*Reade and Collins, 2000*; *Wang et al., 2000*). Our results therefore indicate that the physical coupling between turbulence and copepod density, finite size, and elongated shape does not lead to substantial preferential concentration at the scale of the organisms, at least for the intensity of turbulence and species tested here. They also further confirm the absence of significant clustering observed experimentally in earlier studies with finite-size, neutrally buoyant particles (*Fiabane et al., 2012*).

We note the emergence of heterogeneity at small scales ($r \leq 3$ mm) in the spatial distribution of living copepods swimming in turbulence (*Figure 5*). This phenomenon is not unexpected, since it corresponds to the signature of two organisms moving toward each other at short $r$ within their perception radius. Inward motion increases the time spent by two copepods in close proximity, leading to clustering due to behavior. An interesting observation is that $f(\boldsymbol{x}, \boldsymbol{r})$ is maximal in living copepods swimming in calm water and deviates from one at a further distance (approximately 8 mm) than in turbulence (*Figure 5*). The behavioral mechanisms responsible for enhanced local concentration at larger spatial scales are unknown but are likely to include communication via pheromones trails, which allows males to locate and move toward females beyond their perception radius for hydrodynamic perturbations. Support for the contribution of olfactory orientation in still water comes from the observation of mating events involving pheromone trail-tracking behavior in *E. affinis*, both in earlier studies (*Katona, 1973*) and in our measurements (*Appendix 1—figure 3*).

## Predicting plankton encounter rates in turbulence

Encounter rates in the plankton are governed by several physical and biological processes: motility and transport by flow, which are single-organism properties, and interactions at short range that correlate the behavior of two otherwise independent organisms. Drawing an analogy with the collision of water droplets in clouds (*Falkovich et al., 2002*), the contributions of these processes are captured in the collision kernel, which can itself be decomposed into the product of the geometric collision kernel $K(r)$ and the collision efficiency. The geometric collision kernel $K(r) = \gamma(r)g(r)$ fully defines the collision kernel in the absence of interactions between organisms, and therefore it applies to separation distances above the perception radius. It is a function of $r$ and gives the rate at

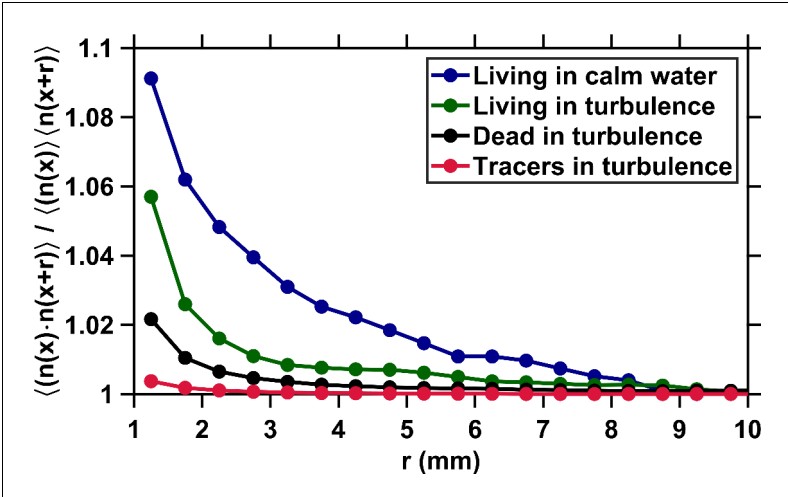

**Figure 5.** The lack of preferential concentration due to particle inertia reduces the geometric collision kernel to the clearance rate. Correlation of the particle concentration versus the voxel separation distance $r$. The brackets denote both time and space (i.e., over $x$) averaging. The minimal separation distance between voxels corresponds to the size of a copepod. The correlation remains very close to one for uniformly distributed tracers (red). It is slightly higher than one at small separations for inert carcasses (black) because of the interactions between small-scale eddies and the density, finite size, and rather elongated shape of the copepods, but the increase in local concentration is negligible when compared to that observed with small and heavy or light particles in turbulence (**Wang et al., 2000**). As a consequence, inertial clustering does not contribute significantly to the encounter rate of copepods in turbulence at any $r$. Behavior results in clustering for living copepods in turbulence (green). This is in good agreement with the increase in $|\langle u_{r\,|\,i}\rangle|$ at $r \leq 4$ mm measured in this study, since this behavior increases the time spent by two copepods in close proximity. However, clustering due to behavior in turbulence is restricted to small $r$ below the radius of hydrodynamic interactions, and therefore does not contribute to the geometric collision kernel for $r$ above 4 mm. Clustering is more significant in copepods swimming in calm water (blue), presumably because of the contribution of chemical communication between adults (**Appendix 1—figure 3**). The online version of this article includes the following source data for figure 5:

**Source data 1.** Source data for **Figure 5**.

which two copepods, moving independently from each other due to motility and physical effects due to turbulence, approach by a distance $r \geq 4$ mm. We have shown in the previous section that the preferential concentration caused by particle inertia and shape is negligible, that is, $g(r) \approx 1$, reducing $K(r)$ to its clearance rate component $\gamma(r)$ that originates from the separate contribution of turbulent advection and the independent self-locomotion of copepods (**Figure 2A**). Drawing from our experimental results, we develop a theory for the encounter rates of motile zooplankton in turbulence for $r$ equal to or above the radius of hydrodynamic interactions. The model takes into account the contributions of turbulence and organism motility and predicts the mean inward pairwise radial relative velocity $\langle u_{r\,|\,i}\rangle$, from which it is possible to estimate $\langle \gamma \rangle$ and therefore $K(r)$ for $r$ equal to or above the perception radius. The model builds on experimental data to derive a general framework that can be applied to other species with intermittent motion. Being semi-empirical, it does not rely on certain assumptions that are common to most existing theoretical formulations, for instance that organisms have a constant ballistic or diffusive motion (**Visser and Kiørboe, 2006**). The model only requires empirical quantities that are easy to measure, ensuring its applicability beyond the species and experimental conditions considered in this work. Another highlight of our approach compared to existing formulations is that it takes into account the contribution of clustering in driving encounters. Indeed, applications to cases where preferential concentration is significant is straightforward via the contribution of $g(r)$ in $K(r)$.

We first consider a pair of copepods swimming in turbulence. The velocity of the first copepod is given by $\boldsymbol{u}_{b,1} + \boldsymbol{u}_t(\boldsymbol{x}_1)$ and the velocity of the second copepod is given by $\boldsymbol{u}_{b,2} + \boldsymbol{u}_t(\boldsymbol{x}_1 + \boldsymbol{r})$, where $\boldsymbol{u}_{b,1}$ and $\boldsymbol{u}_{b,2}$ are the velocity vectors of the organisms with respect to the flow (i.e., corresponding to the behavioral component of their motion), $\boldsymbol{u}_t(\boldsymbol{x}_1)$ is the velocity of the turbulent flow at the position $\boldsymbol{x}_1$

of the first organism, and $r$ is their separation vector. The mean inward pairwise radial relative velocity of the organisms is given by

$$\langle u_{r\,|\,i} \rangle = \langle \left(u_{r,c,b} + u_{r,t}\right)\theta\left(-u_{r,c,b} - u_{r,t}\right)\rangle,$$ (1)

where $u_{r,c,b} = \left(\boldsymbol{u}_{b,2} - \boldsymbol{u}_{b,1}\right)\cdot\hat{r}$ is the behavioral component of the pairwise radial relative velocity of the copepods, $\theta$ is the unit step function, and $u_{r,t} = \left(\boldsymbol{u}_t(\boldsymbol{x}_1 + \boldsymbol{r}) - \boldsymbol{u}_t(\boldsymbol{x}_1)\right)\cdot\hat{r}$ is the pairwise radial relative velocity of the flow. Estimating $\langle u_{r\,|\,i} \rangle$ via *Equation 1* requires knowledge of $u_{r,c,b}$, which is difficult to obtain experimentally as it involves resolving the motion of many organisms swimming simultaneously and that of the underlying flow. However, for separation distances larger than the perception radius, the behavioral component of the copepod velocity $\boldsymbol{u}_b$ can be considered independent (*Figure 2A*), and therefore $\boldsymbol{u}_{b,2} - \boldsymbol{u}_{b,1}$ is independent of $r$, meaning that the average of $\boldsymbol{u}_{b,2} - \boldsymbol{u}_{b,1}$ for a fixed $r$ is independent of the direction of $r$. Because in the ocean turbulence is close to isotropic at small scales, we can with no loss of generality set $\hat{x}$ instead of $\hat{r}$ in *Equation 1*, where $\hat{x}$ is a unit vector in any fixed direction x, and therefore we can write

$$\langle u_{r\,|\,i} \rangle = \int_{-\infty}^{\infty} \langle \left(v_x + u_{r,t}\right)\theta\left(-v_x - u_{r,t}\right)\rangle P(v_x)dv_x,$$ (2)

where $P(v_x)$ is the probability density function of a component of $\boldsymbol{u}_{b,2} - \boldsymbol{u}_{b,1}$ in the direction $\hat{x}$. This can also be written as

$$\langle u_{r\,|\,i} \rangle = \int_{-\infty}^{\infty} \left(v_x + u_{r,t}\right)\theta\left(-v_x - u_{r,t}\right)P(v_x)P(u_{r,t})du_{r,t}dv_x.$$ (3)

$P(v_x)$ in *Equation 3* can be obtained from $P(u_b)$, the probability density function of the magnitude of the velocity of copepods with respect to the flow, which itself can be estimated from $P(u_m)$, the probability density function of their velocity magnitude in calm water. The first advantage of using $P(u_m)$ is that $u_m$ is easily accessible experimentally, meaning that our model can be readily adapted to other species. The second advantage is that it allows estimating $\langle u_{r\,|\,i} \rangle$ for different intensities of turbulence. Indeed, many species of zooplankton, from calanoid copepods to cladocerans, swim by alternating periods of slow cruising motion with frequent relocation jumps. In calanoid copepods, the slow forward motion derives from the creation of feeding currents accomplished by the high-frequency vibration of the cephalic appendages (*Kiørboe et al., 2014*). Relocation jumps originate from the repeated beating of the swimming legs and result in sequences of high velocity bursts leading to an intermittent motion (*Jiang and Kiørboe, 2011*; *Michalec et al., 2017*). We build on this fundamental feature to model $P(u_b)$ as

$$P(u_b) = P_j P_{jumping}(u_m) + (1 - P_j)P_{cruising}(u_m),$$ (4)

where $P_j$ is the fraction of time that a copepod is jumping, $P_{jumping}(u_m)$ is the probability density function of their jump velocities in calm water, and $P_{cruising}(u_m)$ is the probability density function of their cruising velocities in calm water. The model draws on our previous experimental observation that in the calanoid copepod E. affinis, $P_j$ is the only jump-related quantities that varies with the turbulence intensity (*Michalec et al., 2017*). Thus $P_{jumping}(u_m)$ has the same form as in calm water, is determined by $P(u_m)$, and is defined as

$$P_{jumping}(u_m) = \frac{\theta(u_m - v_t)P(u_m)}{\int_{v_t}^{\infty} P(u_m)du_m},$$ (5)

where $v_t \approx 10$ mm s$^{-1}$ is the threshold, determined empirically from $P(u_m)$, that separates cruising velocities from jump velocities (*Appendix 1—figure 4A*). $P_{cruising}(u_m)$ is similarly defined as

$$P_{cruising}(u_m) = \frac{\theta(v_t - u_m)P(u_m)}{\int_{0}^{v_t} P(u_m)du_m}.$$ (6)

The model therefore only requires $P(u_m)$ and $P_j$ to estimate $P(u_b)$ at any hydrodynamic condition (*Appendix 1—figure 4A*). In certain species, $P_j$ is a function of the turbulence intensity (*Michalec et al., 2017*), but in others where the jump frequency may remain constant, we have

$P_j = \int_{v_t}^{\infty} P(u_m)du_m$. We can then derive $P(v_x)$ from $P(u_b)$ as the sum of the probabilities of mutually exclusive events that both organisms jump, one jumps and the other cruises, and both cruise (see Appendix 2 for the complete derivation).

$$P(v_x) = \frac{1}{2}\int_0^{\infty} du_{b,1} \int_0^{u_{b,1}} du_{b,2} \frac{P(u_{b,1})P(u_{b,2})}{u_{b,1}u_{b,2}}$$
$$\times \left[(u_{b,1} + u_{b,2} - |v_x|)\theta(u_{b,1} + u_{b,2} - |v_x|) - (u_{b,1} - u_{b,2} - |v_x|)\theta(u_{b,1} - u_{b,2} - |v_x|)\right]. \tag{7}$$

We show in *Appendix 1—figure 4B* that $P(v_x)$ estimated from *Equation 7* agrees well with the empirical curve. The mean inward pairwise radial relative velocity of copepods swimming in turbulence $\langle u_{r|i} \rangle$ is then obtained from $P(v_x)$ by including the contribution of turbulent velocity differences $u_{r,t}$. We provide here a semi-empirical formula for $P(u_{r,t})$ that is based on our measurements and allows estimating the contribution of turbulent advection over a range of hydrodynamic conditions, but theoretical forms are also available in the literature (*Kailasnath et al., 1992*). We note that in the water column, the turbulence intensity decreases relatively quickly below the surface (*Lozovatsky et al., 2006*; *Sutherland et al., 2013*; *Sutherland and Melville, 2015*). Consequently, the intermittency of turbulence experienced by plankton remains moderate, and the assumption that $u_{r,t}/\epsilon^{1/3}$ has a Reynolds number independent distribution holds with good accuracy for a fixed pairwise separation $r$. We therefore assume that

$$P(u_{r,t}) = \frac{1}{\epsilon^{1/3}}F\left(\frac{u_{r,t}}{\epsilon^{1/3}}\right), \tag{8}$$

where $F(x)$ is a Reynolds number independent function and $x = u_{r,t}/\epsilon^{1/3}$. We find from our data that $F(x)$ can be fitted with stretched exponentials (*Appendix 1—figure 4B*), which is expected since they provide good approximation to $P(u_{r,t})$ (*Kailasnath et al., 1992*), thereby allowing us to estimate $P(u_{r,t})$ as a function of $\epsilon$. $\langle u_{r|i} \rangle$ is then given from *Equation 3* and *Equation 8* by

$$\langle u_{r|i} \rangle = \int_{-\infty}^{\infty} \left(v_x + \epsilon^{1/3}x\right)\theta\left(-v_x - \epsilon^{1/3}x\right)P(v_x)F(x)dx\,dv_x. \tag{9}$$

Given the simplicity of the model, the agreement between the estimated values and experimental data is satisfactory. For a pairwise separation distance $r = 4$ mm, we estimate $\langle \gamma \rangle = 1.2$ cm$^3$ s$^{-1}$ via *Equation 9*, versus $\langle \gamma \rangle = 1.6$ cm$^3$ s$^{-1}$ from the measurements. An interesting point is that $\langle \gamma \rangle$ is dominated by the contribution of jumps. The two cases in *Equation 7* where at least one organism of the pair jumps, that is, when both jump or one jumps and the other cruises, contribute more to $|\langle u_{r|i} \rangle|$ than the case where both organisms are cruising (approximately 4.7 mm s$^{-1}$ versus 2.7 mm s$^{-1}$, respectively). This result highlights the importance of jumps in sustaining efficient mate finding in turbulence. It also confirms that the mechanism underpinning higher encounter rates at separation distances comparable to the radius of hydrodynamic interactions is the independent and vigorous self-locomotion of individual organisms. Although the model slightly underestimates encounter rates because of the difficulties of estimating $P(u_b)$ in turbulence from $P_{jumping}(u_m)$ in calm water (*Appendix 1—figure 4A*), the measured and estimated values are close, highlighting the robustness of our approach and indicating that combining turbulent advection with the independent motion of individual organisms correctly predicts encounter rates at separation distances corresponding to the radius of hydrodynamic interactions. Previous estimations of plankton encounter rates in turbulence are based on theoretical formulations that often require adjusting free parameters (*Pécseli et al., 2014*), whereas our model is semi-empirical. It also avoids relying on simplified assumptions, for instance on the movement patterns of the organisms (*Visser and Kiørboe, 2006*). It requires only knowledge of the probability density function of the velocity in calm water, which is readily accessible experimentally. Therefore, we expect our model to remain valid for organisms displaying a wide range of motility patterns, allowing accurate predictions when studying mating, predation, and resource exploitation in the plankton.

## Conclusion
Our results identify the physical and behavioral mechanisms that enable calanoid copepods, tiny crustaceans that dominate the zooplankton biomass and represent the most abundant metazoans in

the ocean and estuaries, to maintain efficient mate finding when ambient flow challenges their limited swimming abilities and impairs motion strategies and olfactory orientation. We have previously shown that certain copepods increase their swimming activity in turbulence by performing more frequent relocation jumps that appear uncorrelated to localized flow signals and that persist in time (*Michalec et al., 2017*). While jumping, they reach velocities that are larger than the turbulent velocity in typical oceanic conditions (*Yamazaki and Squires, 1996*). The ecological significance of these jumps remains unknown. A more vigorous motility in turbulence may allow copepods to depart from the underlying flow streamlines, to transition from being passively transported by the flow to being able of directed motion, and to navigate in the water column in spite of the physical constraints that turbulence imposes on their motion (*Genin et al., 2005*). Swimming vigorously may also permit copepods to retain the fitness advantages provided by motility in terms of inter-individual interactions.

We study this important aspect of plankton ecology in the context of mating and show that self-locomotion indeed creates large differences in the inward component of the pairwise radial relative velocity of copepods. This directly results in high encounter rates at separation distances comparable to the spatial extend of the flow field they generate while swimming (*Kiørboe et al., 2014*). A second mechanism, mediated by males only and consisting of directed motion toward neighbor organisms within the perception radius, allows copepods to convert the high encounter rate provided by active motion in turbulence into actual contact events within the short time frame of the encounter. Rheotaxis toward nearby organisms within the perception radius occurs both in calm water and in turbulence, which reveals the ability of copepods to correctly identify the flow signals generated by a conspecific amid the background noise of turbulence. The combination of these two behavioral mechanisms with turbulent advection results in an encounter rate that is substantially larger than that resulting from passive transport in turbulence, as evidenced by a clearance rate ratio $\langle \gamma_{l,t} \rangle / \langle \gamma_{d,t} \rangle \approx 3$ at short separations (*Figure 2B*). The encounter rate in turbulence is also comparable to or even larger than that achieved by active motion in calm hydrodynamic conditions, where swimming strategies and olfactory orientation are possible. We note that the large encounter rate of copepods swimming in turbulence originates primarily from their active motion and not from the contribution of turbulence. The large ratio $\langle \gamma_{l,t} \rangle / \langle \gamma_{tra} \rangle$ indicates that turbulent velocity differences contribute only marginally to encounter rates compared to organism motility (*Figure 2B*). The low ratio $\langle \gamma_{d,t} \rangle / \langle \gamma_{l,c} \rangle$ at small $r$ reveals that turbulent advection, when considered alone, results in a substantially lower encounter rate compared to self-locomotion in calm water (*Figure 2B*). This indicates that being passively transported by the flow does not increase encounter rates beyond those achieved by motility in calm hydrodynamic conditions for organisms and turbulent velocities comparable to those considered in this work (*Rothschild and Osborn, 1988*; *Visser et al., 2009*). It requires active swimming for copepods to achieve an encounter rate that is comparable to or even larger than in calm water.

These results are significant because they provide empirical evidence that two key conditions for mating in turbulence are met. Firstly, the combination of active swimming and turbulent velocity differences increases the flux of organisms within the perception radius, thereby enhancing encounter rates. This mechanism has been extensively studied and is now well established in oceanography (*MacKenzie et al., 1994*; *Lewis and Pedley, 2000*). Secondly, copepods are able to convert high encounter rates to contact events by directed motion within the short time frame of the encounter. By manually extending the trajectories of the two individuals of the pair after the time of contact, we provide evidence that mating occurs after contact is made (*Figure 4*). Theoretical studies on the capture success of larval fish feeding on zooplankton indicate that strong turbulence reduces feeding rates below those achieved in calmer conditions because of a decrease in the probability of successful pursuit once an encounter has occurred (*MacKenzie et al., 1994*; *Kiørboe and MacKenzie, 1995*). These pioneering studies, together with conclusive evidences for mate-tracking behavior from measurements conducted in calm water, have shaped our present understanding of copepod mating (*Doall et al., 1998*; *Yen et al., 1998*; *Bagøien and Kiørboe, 2005*). It has long been assumed that copepods are evolved to mate in relatively quiet waters, although this was never confirmed. Our empirical results add to our knowledge on this fundamental issue. They suggest that reproduction is not restricted to temporal and spatial windows of calm hydrodynamic conditions, such as close to the pycnocline in the stratified ocean or during the tidal current reversal associated

with the absence of flow in estuaries, where motility strategies and olfactory orientation that have been observed in calm water may remain significant and where the effect of turbulence on pursuit success is not strong enough to negate the increase in encounter rates. The ability for efficient mate finding even in relatively strong turbulence relaxes the boundaries between quiescent and turbulent conditions in terms of plankton fitness. It suggests that copepods have adapted to the heterogeneous hydrodynamic conditions that they experience in their dynamic environment, where $\epsilon$ can vary over six orders of magnitude (*Yamazaki and Squires, 1996*). This finding has large implications for our understanding of copepod ecology and of the relationships between turbulence intensity in the environment and the fitness of copepod populations.

Developing realistic models of plankton encounter rates in turbulence is challenging because of the complex coupling between flow and behavior (*Boffetta et al., 2006*; *Pécseli et al., 2010*; *Ardeshiri et al., 2017*) and because the validity of theoretical approaches has seldom been tested empirically due to the difficulties in tracking organisms swimming and interacting in three dimensions and in turbulence. The first model (*Rothschild and Osborn, 1988*) derives from a formulation that considers organisms swimming independently in random direction and in calm water (*Gerritsen and Strickler, 1977*). It introduces the contribution of turbulence to encounter rates by superposing squared velocity differences on top of organism motion (*Rothschild and Osborn, 1988*). However, particles and plankton move in the same coherent small-scale eddy at short separation distances corresponding to the radius of perception. A number of investigations have suggested improvements to this pioneering theoretical framework. In particular, they have considered the correlation of the flow velocity as the separation distance decreases, as well as the influence of the perception capabilities of the organisms (*Kiørboe and MacKenzie, 1995*; *Lewis and Pedley, 2000*; *Pécseli et al., 2014*). These models have allowed oceanographers to study the implications of turbulence-enhanced contact rates on grazing, predation, and optimal strategies in the plankton (*MacKenzie et al., 1994*; *Lewis and Pedley, 2001*; *Lewis, 2003*; *Visser et al., 2009*; *Pécseli et al., 2019*) and they often show good agreement between analytical results and field or laboratory data (*Pécseli et al., 2019*). In this work, we provide a complete semi-empirical formulation for the geometric collision kernel $K(r)$ for $r$ above or equal to the perception radius that shows satisfactory quantitative agreement with our empirical data. The model deviates from previous formulations in that it uses as a starting point the probability density function of the magnitude of the velocity of the organisms in still fluid, from which it is possible to recover their pairwise radial relative velocity in turbulence. It does not rely on simplifying assumptions that are common in encounter rate models, for instance that the organisms move by cruising or by alternating pauses and periods of active swimming (*MacKenzie et al., 1994*; *Kiørboe and MacKenzie, 1995*). It also accounts for the effects of turbulence on behavior. Indeed, certain species of zooplankton react to specific flow signals in their environment. They can sense and respond to hydrodynamic forces by modulating their swimming activity and velocity (*Michalec et al., 2017*; *Fuchs et al., 2018*). Their interactions with turbulence therefore include active behavioral responses in addition to physical effects due to the forces exerted by the flow. The model incorporates these indirect effects via the parameter $P_j$, the fraction of time that a copepod is jumping. Therefore, our approach allows predicting encounter rates for different species that have different velocity distributions and different responses to turbulence. The model draws an analogy with the collision of water droplets in clouds. It considers the contribution of motility and turbulence in the pairwise radial relative velocity between copepods to predict the flux of organisms within the perception radius. It allows accounting for clustering effects due to particle density, shape, and finite size, since applications to cases where preferential concentration is significant is straightforward via the contribution of $g(r)$ in $K(r)$.

The second term of the collision kernel, termed the collision efficiency, provides corrections to $K(r)$ that account for interactions between organisms at $r$ below the range of validity of $K(r)$, that is, for $r \leq 4$ mm within the clearance volume. From a modeling perspective, an exhaustive estimation of the collision kernel can be achieved by multiplying the rate at which two organisms cross their radius of hydrodynamic interactions, given by $K(r = 4$ mm$)$, with the probability of successful contact following inward motion at shorter range, captured in the collision efficiency. It was however not possible to quantify the collision efficiency from our measurements at the same degree of accuracy because of the difficulties of resolving interactions between organisms at very small spatial scale while recording their motion in a much larger volume. Foreseen detailed observations within the radius of hydrodynamic interaction will allow quantifying the ability of male copepods to make

contact with females when both the velocity gradients of turbulence and the escape reaction of females may oppose their inward motion. They will also provide additional information on the impacts of turbulence on the perception capabilities of copepods and on their consequences on the perception radius and therefore encounter rates. Very little information is currently available on this important issue (*Lewis, 2003*; *Pécseli and Trulsen, 2016*). However, we show here that the perception distance is similar in calm water and in turbulence, as suggested by males moving inward, starting at the same separation distance ($r \approx 4$ mm). Therefore, the perception capability appears to be relatively unaffected for this species, and we can already anticipate that the encounter rate will not vary substantially.

A fundamental determinant of the life of plankton is the interplay between organism behavior and turbulence, a prevalent feature of their environment. Our observation of a physical and behavioral coupling mechanism that sustains efficient mate finding in copepods amid ambient fluid motion, together with recent reports on the behavioral adaptations evolved by other zooplankton to enhance survival in turbulence (*DiBacco et al., 2011*; *Fuchs et al., 2018*), illustrates the implications of this coupling in terms of organism fitness and how it influences or even governs important biological processes at the base of the trophic network. We suggest that the ability of copepods to find mates in turbulence enables them to thrive in energetic ecosystems and has contributed to their formidable evolutionary success and widespread distribution in marine, coastal, and estuarine environments.

## Materials and methods

### Plankton cultures

Our model species is the calanoid copepod *Eurytemora affinis*. This species complex includes a number of genetically divergent but morphologically similar populations inhabiting estuaries, lakes, salt marshes, and the Baltic Sea. It often dominates the mesozooplankton community in the low-to-medium salinity zone of most temperate estuaries, where it represents an important component of the food web (*Mouny and Dauvin, 2002*; *Kimmel and Roman, 2004*; *Devreker et al., 2010*). We used organisms from our plankton rearing facility at Lille University. The cultures originate from individuals sampled in September 2014 from the oligohaline zone of the Seine Estuary (France). Copepods were grown in aerated 300 L containers at a temperature of 18°C, at salinity 15 (sterilized seawater from the English Channel adjusted to salinity with deionized water), and under a fluorescent light cycle of 12L:12D. They were fed with the micro-algae *Rhodomonas baltica* and *Isochrisis galbana* cultured in autoclaved seawater at salinity 30, in Conway medium, under a 12L:12D light cycle, and at a temperature of 18°C. Copepods and algae were collected from the stock cultures and shipped overnight to ETH Zurich in refrigerated containers. Measurements were conducted in October 2017, within a few days after shipment, and after acclimation of the copepods to the new laboratory conditions.

### Experimental setup

We recorded the motion of copepods and flow tracers by means of three-dimensional particle tracking velocimetry. This technique identifies and follows individual particles in time and provides a Lagrangian description of their motion in three dimensions. It was originally developed to measure velocity and velocity gradients along tracer trajectories in turbulent flows (*Maas et al., 1993*; *Malik et al., 1993*; *Lüthi, 2002*), and we have previously applied it to study the swimming behavior of small aquatic organisms (*Michalec et al., 2017*; *Sidler et al., 2017*). The recording system was composed of four synchronized Mikrotron EoSens cameras. Three cameras were equipped with red band-pass filters and recorded the motion of fluorescent tracer particles from one side of the aquarium. The fourth camera was equipped with a green band-pass filter and mounted in front of an image splitter, which is an optical arrangement that allows stereoscopic imaging using one single camera. This camera recorded the motion of copepods from the opposite side of the aquarium. The cameras were fitted with Nikon 60 mm lenses and recorded on two *DVR Express Core 2* devices (IO Industries) at 200 Hz and at a resolution of 1280 by 1024 pixels. Illumination was provided by a pulsed laser (wavelength of 527 nm, pulse energy of 60 mJ) operating at 5 KHz. The aquarium was 27 cm (width) by 18 cm (depth) by 17 cm (height) and contained a forcing device creating

homogeneous and isotropic turbulence (*Hoyer et al., 2005*). The flow was forced by two arrays of four counter-rotating disks located on the lateral sides of the aquarium. The disks were 40 mm in diameter and smooth to prevent mechanical damage to the copepods. They were driven by a servo-motor through a fixed gear chain mounted on top of the aquarium and they rotated at the same rate.

### Recording conditions

We restricted our measurements to a 6 cm (height) by 6 cm (width) by 2 cm (depth) volume centered in the middle of the aquarium, midway between the disks, where the turbulence is almost homogeneous and isotropic (*Hoyer et al., 2005*). To record the motion of the flow, we used fluorescent tracer particles with a material density $\rho_p$ = 1.01 g cm$^{-3}$ and a mean diameter $d_p$ = 69 μm. The Stokes number of particles smaller than the Kolmogorov length scale $\eta$ is given by $St = (1/18)(\rho_p/\rho_f)(d_p/\eta)^2$ where $\rho_f$ is the density of the fluid. In our measurements, $St = 10^{-3}$, indicating that these particles behave as passive tracers. The mean distance between tracers was approximately 1.5 mm, yielding a spatial resolution of about 3 $\eta$. We checked copepods for integrity under a microscope and selected healthy individuals only. For each measurement, we transferred copepods (size fraction above 300 μm, corresponding to adults and late-stage copepodites) into the aquarium, and allowed them to acclimate for 30 min. The fraction of males, females and copepodites was made roughly similar during the sorting. The number density was approximately one individual per cubic cm. This density is on the upper range of values observed in the field (*Devreker et al., 2008*). This allowed to record many copepods in the investigation volume and to obtain statistically robust quantities even at short separation distances. We recorded the motion of copepods and tracer particles in still water and in turbulence. For each condition, we conducted two measurements, using new individuals from the cultures. Each sequence lasted 5 min. The sequences in turbulence were preceded by an additional minute with the disks spinning, for the copepods to acclimate to turbulence and for the flow to reach statistical stationarity. Water temperature increased from 18°C to 19°C at the end of the recording. We conducted the same measurements using dead copepods to account for the effects of particle size and density. We obtained 10,643,329 and 7,535,728 data points for living copepods in calm water and turbulence, respectively, and 18,606,682 data points for inert carcasses in turbulence.

### Particle tracking and trajectory processing

We calibrated the cameras using a calibration block on which reference points of known coordinates are evenly distributed along the three directions, and performed an additional dynamic calibration based on the images of moving particles (*Liberzon et al., 2012*). Knowing the camera intrinsic and extrinsic parameters, we established correspondences between particle image coordinates and retrieved the three-dimensional positions of the moving particles by forward intersection. We tracked tracers and copepods using an algorithm based on image and object space information (*Willneff, 2003*). We connected broken trajectory segments by applying a predictive algorithm that uses the position, velocity, and acceleration of the particles along their trajectories (*Michalec et al., 2017*). Trajectories were smoothed with a third-order polynomial filter, and the velocity of the particles was directly estimated from the coefficients of the polynomial. The width of the filter was set to 21 points, corresponding to approximately half of the Kolmogorov time scale. This value represents a good compromise between improving the measurement of the velocity and preserving the features of the data, especially the strong velocity fluctuations that result from the intermittent motion of copepods (*Michalec et al., 2015b*). To express the coordinates of the copepods in the reference frame of the tracer particles, we registered a set of tracer trajectories recorded from the two sides of the aquarium, and obtained the three-dimensional rigid transformation that aligned the two coordinate systems.

### Flow parameters

We used the velocity of tracers to interpolate the flow velocity at the position of the copepods and at each time step. The interpolation procedure uses weighted contributions from nearby tracers according to their separation distance from the copepod. Nearby tracers are defined as tracers found within a sphere of 5 mm radius centered at the location of the copepod. Although this radius

is larger than the Kolmogorov length scale $\eta$, the velocity was resolved with sufficient accuracy: we obtained a relative error of 8 % for the velocity gradient tensor, using kinematic checks based on the acceleration and on the incompressibility of the velocity field (*Lüthi et al., 2005*). We estimated the space- and time-averaged energy dissipation rate $\epsilon$ in the investigation volume from the relation $\langle \delta_r \boldsymbol{u} \cdot \delta_r \boldsymbol{a} \rangle = -2\epsilon$, where $\langle \delta_r \boldsymbol{u} \cdot \delta_r \boldsymbol{a} \rangle$ is the velocity-acceleration structure function, $\delta_r$ denotes the Eulerian spatial increment of a quantity with respect to the pairwise separation distance $r$, and $\boldsymbol{u}$ and $\boldsymbol{a}$ are the velocity and acceleration vectors of tracers, respectively (*Ott and Mann, 2000*). This estimate was compared to the relation $\epsilon \simeq C_\epsilon u_{rms}^3 / L$ where $C_\epsilon$ is a coefficient of order one, $u_{rms}$ is the root-mean-square of the velocity fluctuations of tracers, and $L$ is the integral length scale of the flow, estimated for each experimental condition via the Eulerian velocity autocorrelation function. Both methods yielded comparable results.

## Acknowledgements

This work was supported by grant No. 172916 from the Swiss National Science Foundation. We thank the *Communauté d'Agglomération du Boulonnais* and Lille University for supporting the implementation of a large-scale copepod rearing pilot project, and the past and current members of the group of SS for maintaining the continuous plankton cultures.

## Additional information

### Funding

| Funder | Grant reference number | Author |
| --- | --- | --- |
| Swiss National Science Foundation | 172916 | Markus Holzner |

The funders had no role in study design, data collection and interpretation, or the decision to submit the work for publication.

### Author contributions

François-Gaël Michalec, Conceptualization, Software, Formal analysis, Investigation, Visualization, Methodology, Writing - original draft, Designed experiments. Performed experiments, Analyzed the data, Wrote the manuscript; Itzhak Fouxon, Methodology, Writing - review and editing, Developed the model and helped in analyzing the data, Helped draft the manuscript; Sami Souissi, Resources, Writing - review and editing, Provided the test organisms, Helped drafting the manuscript; Markus Holzner, Resources, Supervision, Funding acquisition, Project administration, Writing - review and editing, Designed experiments, Helped drafting the manuscript

### Author ORCIDs

François-Gaël Michalec https://orcid.org/0000-0002-4232-0665

### Decision letter and Author response

Decision letter https://doi.org/10.7554/eLife.62014.sa1
Author response https://doi.org/10.7554/eLife.62014.sa2

## Additional files

### Supplementary files

• Transparent reporting form

### Data availability

All data generated or analysed during this study are included in the manuscript and supporting files. Source data files have been provided for Figures 2, 3 and 5.

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

## Appendix 1

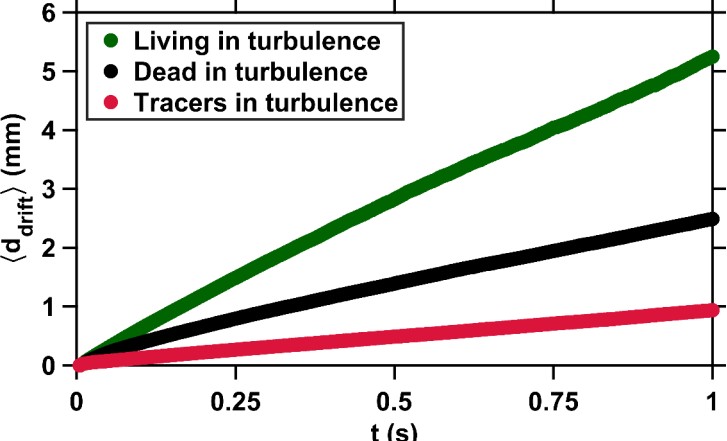

**Appendix 1—figure 1.** Copepods swimming in turbulence depart from the flow streamlines because of their slight inertia and active motion. Average drift distance $\langle d_{drift}\rangle$ versus time to encounter $t$. The small drift of tracers (red) is due to the propagation in time of small uncertainties in the computation of the flow velocity at the particle location. These uncertainties have been estimated at approximately 8 % for the velocity gradient tensor, based on kinematic checks (**Lüthi et al., 2005**). The deviation of dead copepods (black) from the flow streamlines results from inertial effects caused by their finite size, elongated shape, and density slightly larger than that of the carrier fluid. The much larger deviation of living copepods (green) is caused by the cumulative effects of inertia and active motion via frequent relocation jumps (**Michalec et al., 2017**). $\langle d_{drift}\rangle$ has been computed from several hundreds of encounter events for each case (living copepods, inert carcasses, and tracers).

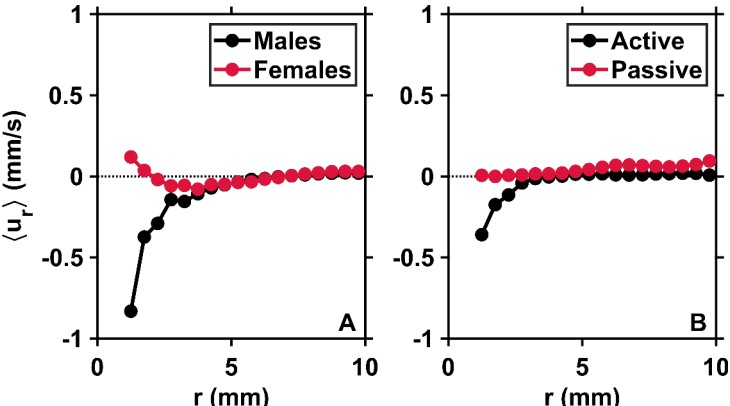

**Appendix 1—figure 2.** Males initiate inward motion at short separations. (**A**) Mean pairwise radial relative velocity $\langle u_r\rangle$ versus the pairwise separation distance $r$ for males (black) and females (red) swimming separately in calm water. Negative values of $\langle u_r\rangle$ at short $r$ for males indicate attraction. Positive values of $\langle u_r\rangle$ for females indicate repulsion. These results originate from complementary measurements conducted with single genders of adult copepods. (**B**) Mean radial relative velocity $\langle u_r\rangle$ versus the pairwise separation distance $r$ for males and females swimming together in turbulence in approximately equal proportions. $\langle u_r\rangle_{active}$ is computed using the velocity of the organisms with respect to the flow. It quantifies the behavioral component of the motion. $\langle u_r\rangle_{passive}$ is computed using the instantaneous flow velocity at the location of the copepods. It quantifies the component of the motion that results from turbulent advection. The net inward flux is smaller in the mixed-gender measurements because of the contribution of females, but still clearly visible because $|\langle u_r\rangle|_{\male} > |\langle u_r\rangle|_{\female}$ at short $r$.

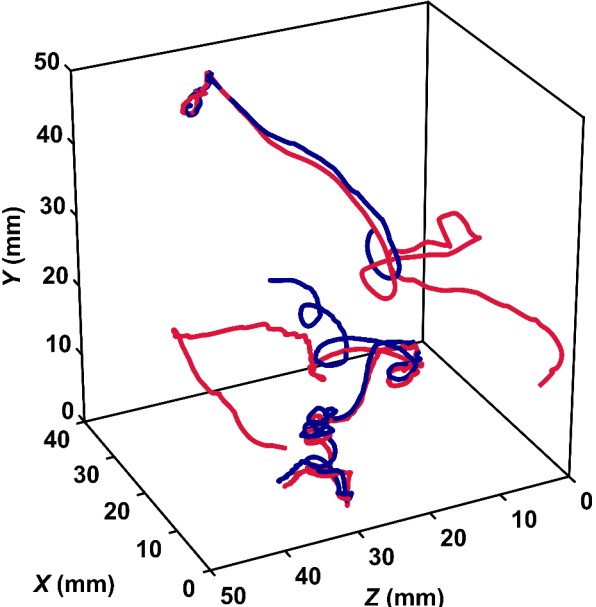

**Appendix 1—figure 3.** Pheromone trails facilitate mating encounters in calm water. Two examples of mating interactions mediated by pheromones in the calanoid copepod *Eurytemora affinis*. This species is well known for using pheromones for mate location (*Katona, 1973*). A male (red) detects the pheromone trail released by a female (blue) and swims up the pheromone gradient with great accuracy until contact. The distance between the male and the female during the pursuit is larger than the radius of perception for hydrodynamic interactions (*Bagøien and Kiørboe, 2005*).

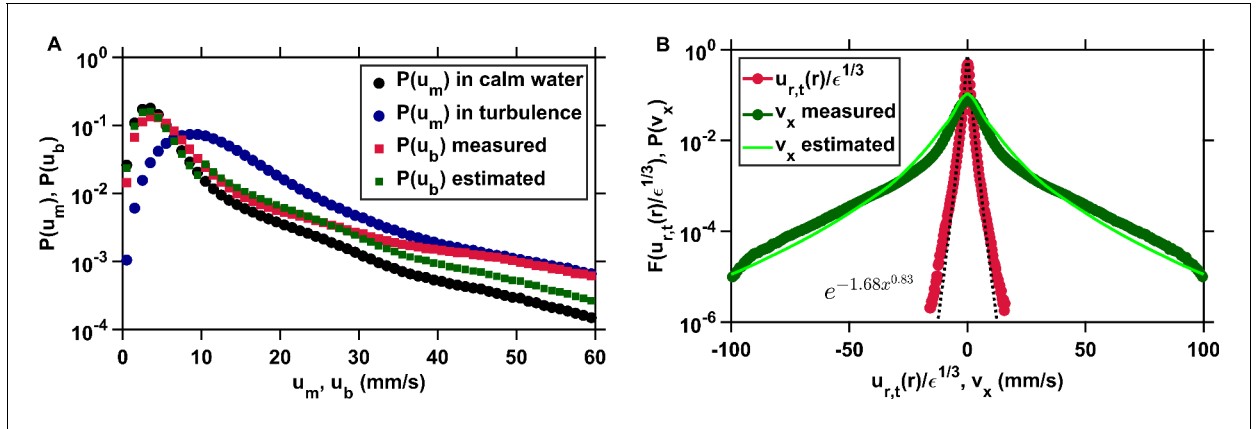

**Appendix 1—figure 4.** The clearance rate $\langle\gamma\rangle$ of copepods in turbulence, and therefore their encounter rate, can be estimated down to the radius of hydrodynamic interactions from the probability distributions of the pairwise radial relative velocity of turbulence and of the organism velocity. (A) Probability density functions $P(u_m)$ of the velocity of copepods swimming in calm water (black, circles) and in turbulence (blue, circles), and probability density functions of the velocity of copepods with respect to the turbulent flow $P(u_b)$ from the experimental data (red, squares) and estimated from $P_{jumping}(u_m)$ and $P_{cruising}(u_m)$ in calm water (green, squares), using $v_t \approx 10$ mm s$^{-1}$ as the threshold, determined empirically from $P(u_m)$, that separates cruising velocities from jump velocities. $P(u_b)$ in turbulence is approximated from $P(u_m)$ in calm water, which is much simpler to obtain experimentally. (B) From $P(u_b)$ it is possible to derive $P(v_x)$, the probability density function of one component of the velocity differences of two copepods, using their velocity with respect to the flow, which is not trivial to obtain empirically. The agreement between $P(v_x)$ from the data (circles, dark green) and estimated from $P(u_b)$ (solid line, light green) is good. The mean inward radial velocity of copepods in turbulence, and therefore $\langle\gamma\rangle$, is then obtained by adding the contribution of flow motion, given by the probability density function $F(u_{r,t}(r)/\epsilon^{1/3})$ (circles, red) where $\epsilon$ is the dissipation rate of the turbulent kinetic energy and $u_{r,t}(r)$ is the pairwise radial relative velocity of tracers at separation $r = 4$ mm corresponding to the radius of hydrodynamic interactions (*Bagøien and Kiørboe, 2005*). Stretched exponentials (dashed line, black) provide good approximation to $F(u_{r,t}(r)/\epsilon^{1/3})$.

## Appendix 2

### Derivation of $P(v_x)$ from $P(u_b)$

In the following, we derive $P(v_x)$, a quantity that is difficult to obtain experimentally, from $P(u_b)$, which was previously derived from $P(u_m)$, the probability density function of the velocity of copepods in calm water. The velocity of the first copepod is given by $\boldsymbol{u}_{b,1} + \boldsymbol{u}_t(\boldsymbol{x}_1)$ and the velocity of the second copepod by $\boldsymbol{u}_{b,2} + \boldsymbol{u}_t(\boldsymbol{x}_1 + \boldsymbol{r})$, where $\boldsymbol{u}_{b,1}$ and $\boldsymbol{u}_{b,2}$ are the velocity vectors of the organisms with respect to the flow (i.e., corresponding to the behavioral component of their motion), $\boldsymbol{u}_t(\boldsymbol{x}_1)$ is the velocity of the turbulent flow at the position $\boldsymbol{x}_1$ of the first organism, and $\boldsymbol{r}$ is their separation vector. We introduce $\boldsymbol{v} = \boldsymbol{u}_{b,2} - \boldsymbol{u}_{b,1}$ and its probability density function $P(\boldsymbol{v})$. We assume that $P(\boldsymbol{v})$ depends on $v = \|\boldsymbol{v}\|$ only because of isotropy, so that $P(\boldsymbol{v}) = P(v)$. $P(v_x)$, the probability density function of one component of $\boldsymbol{v}$, is then given in terms of $P(v)$ by

$$P(v_x) = \langle \delta(v\cos\theta - v_x) \rangle = 2\pi \int_0^\infty v^2 P(v)dv \int_{-1}^1 dx\, \delta(vx - v_x) = 2\pi \int_{|v_x|}^\infty v P(v)dv, \tag{A1}$$

where $x = \cos\theta$ is the cosine of the polar angle $\theta$ in spherical coordinates. We can readily check the normalization,

$$\int_{-\infty}^\infty P(v_x)dv_x = 4\pi \int_0^\infty v P(v)dv \int_0^v dv_x = 1. \tag{A2}$$

We derive $P(v)$ from $P(\boldsymbol{u}_b)$, the probability density function of the three components of the copepod velocity with respect to the flow, which can be obtained in terms of $P(u_b)$, the probability density function of $\|\boldsymbol{u}_b\|$.

$$P(\boldsymbol{u}_b) = \frac{P(u_b)}{4\pi v^2}, \quad \int_0^\infty P(u_b)du_b = 1. \tag{A3}$$

Because the two vectors $\boldsymbol{u}_{b,1}$ and $\boldsymbol{u}_{b,2}$ are independent from each other for separation distances above the radius of hydrodynamic interaction, and because the distributions of $-u_{b,1}$ and $u_{b,1}$ coincide, we can compute the characteristic function of $\boldsymbol{v}$ as the square of $P(k)$, the Fourier transform of $P(\boldsymbol{u}_b)$. Assuming again that $P(\boldsymbol{u}_b) = P(u_b)$ because of isotropy, $P(k)$ is given in terms of $P(u_b)$ by

$$
\begin{aligned}
P(k) &= \int \exp(-i\boldsymbol{k}\cdot\boldsymbol{u}_b)P(\boldsymbol{u}_b)d\boldsymbol{u}_b \\
&= 2\pi \int_0^\infty u_b^2 P(u_b)du_b \int_{-1}^1 \exp(iku_b x)dx \\
&= 4\pi \int_0^\infty \frac{u_b P(u_b)\sin(ku_b)du_b}{k} \\
&= \int_0^\infty \frac{P(u_b)\sin(ku_b)du_b}{ku_b} \\
&= \frac{1}{2}\mathrm{Im}\left( \int_{-\infty}^\infty \frac{P(|u_b|)\exp(iku_b)du_b}{ku_b} \right) \\
&= -\frac{1}{2k}\frac{d}{dk}\int_{-\infty}^\infty \frac{P(u_b)\exp(iku_b)du_b}{u_b^2} \\
&= -\frac{2\pi}{k}\frac{d}{dk}\int_{-\infty}^\infty P(u_b)\exp(iku_b)du_b,
\end{aligned}
\tag{A4}
$$

where the last line is real. We can now retrieve $P(v)$ from $P^2(k)$, the characteristic function of $\boldsymbol{v}$.

$$
\begin{aligned}
P(v) \;&=\; \int \frac{d\boldsymbol{k}}{(2\pi)^3} \exp(i\boldsymbol{k}\cdot\boldsymbol{v}) P^2(k) \int_0^\infty \frac{k^2 P^2(k) dk}{(2\pi)^2} \int_{-1}^1 \exp(ikvx)dx \\
&=\; \int_0^\infty \frac{kP^2(k)\sin(kv)dk}{2\pi^2 v} \\
&=\; \frac{1}{2}\mathrm{Im}\left( \int_{-\infty}^\infty \frac{kP^2(|k|)\exp(ikv)dk}{2\pi^2 v} \right) \\
&=\; -\frac{1}{2v}\frac{d}{dv}\int_{-\infty}^\infty \frac{P^2(|k|)\exp(ikv)dk}{2\pi^2}.
\end{aligned}
\tag{A5}
$$

Using **Equation A1**, we obtain $P(v_x)$ from $P(v)$ and therefore from $P^2(k)$ as

$$
P(v_x) = -\frac{1}{2\pi}\int_{|v_x|}^\infty dv \frac{d}{dv} \int_{-\infty}^\infty P^2(|k|)\exp(ikv)dk = \int_{-\infty}^\infty P^2(|k|)\exp(ik|v_x|)\frac{dk}{2\pi}.
\tag{A6}
$$

Using the formula for $P(k)$ as a function of $P(u_b)$ in **Equation A4**, this can be written as

$$
P(v_x) = \int_0^\infty du_{b,1} du_{b,2} \frac{P(u_{b,1})P(u_{b,2})}{u_{b,1}u_{b,2}} \int_{-\infty}^\infty \exp(ik|v_x|)\frac{\sin(ku_{b,1})\sin(ku_{b,2})}{k^2}\frac{dk}{2\pi}.
\tag{A7}
$$

We use that

$$
\begin{aligned}
\int_{-\infty}^\infty \exp(ik|v_x|)\frac{\sin(ku_{b,1})\sin(ku_{b,2})}{k^2}\frac{dk}{2\pi} \;&=\; \int_{-\infty}^\infty \exp(ik|v_x|)\frac{1-\cos(k(v_1+v_2))}{k^2}\frac{dk}{4\pi} \\
&=\; -\int_{-\infty}^\infty \exp(ik|v_x|)\frac{1-\cos(k(v_1-v_2))}{k^2}\frac{dk}{4\pi},
\end{aligned}
\tag{A8}
$$

and that (**Bateman Manuscript Project, 1954**)

$$
\int_0^\infty \frac{\cos(k|v_x|)(1-\cos(ak))dk}{k^2} = \frac{\pi(|a|-|v_x|)\theta(|a|-|v_x|)}{2},
\tag{A9}
$$

to obtain $P(v_x)$ as a function of $P(u_b)$,

$$
\begin{aligned}
P(v_x) \;=\; &\frac{1}{2}\int_0^\infty du_{b,1}\int_0^{u_{b,1}} du_{b,2} \frac{P(u_{b,1})P(u_{b,2})}{u_{b,1}u_{b,2}} \\
&\times \left[(u_{b,1}+u_{b,2}-|v_x|)\theta(u_{b,1}+u_{b,2}-|v_x|) - (u_{b,1}-u_{b,2}-|v_x|)\theta(u_{b,1}-u_{b,2}-|v_x|)\right].
\end{aligned}
\tag{A10}
$$

