## [Decision Letter]

**Acceptance summary:**

The manuscript addresses the ability of copepods to find mates in a turbulent environment, a major question in marine science. Through comprehensive experiments and models the authors demonstrate that mating occurs despite turbulence and that turbulence facilitates encounters. Animals are brought close to one another by the underlying flow; when a male perceives a potential mate, it achieves contact by direct and vigorous swimming. Perception occurs within 4 mm from the potential mate, and, intriguingly, this range is similar in calm and turbulent environments, suggesting copepods sensory systems are insensitive to turbulence.

**Decision letter after peer review:**

Thank you for submitting your work entitled "Efficient mate finding in planktonic copepods swimming in turbulence" for consideration by *eLife*. Your article has been reviewed by two peer reviewers, and the evaluation has been overseen by a Reviewing Editor and a Senior Editor. The reviewers have opted to remain anonymous.

The manuscript addresses the ability of copepods to find mates in a turbulent environment, a major question in marine science. The authors investigate the encounter rates by experimentally monitoring alive and dead copepods in a turbulent flow and model their individual trajectories. These are interesting data and a robust modelling methodology which merits publication, however we find that the level of biological insight provided by this work is not sufficient to grant publication in *eLife*. There is a long literature about encounter rates of copepods in turbulent environment, including by these authors, and the overall picture is largely consistent with the author's findings. The main novelty of the present manuscript is the focus on mating.

However, the authors do not currently monitor actual mating events.

Documenting mating is especially important because it is not clear whether these animals do mate at such high turbulence intensity. In fact, trace following behavior by males suggests these animals may be adapted to mate at low turbulence levels (although this remains to be proven).

If the authors were able to distinguish males from females, the experiments may be adapted to documenting the actual mating between individuals. We would welcome such a study that fills a void in current knowledge, and would consider it as a new submission as this entirely changes the focus of the manuscript.

Reviewer #1:

This manuscript addresses an important aspect of marine science. Copepods are the most abundant metazoan on the planet and play an important role in recycling organic matter and in passing organic matter to higher trophic levels. The aim of this paper was to model the encounter rates of copepods in a turbulent environment and test the model by observing the interaction of copepods (Eurytemora affinis) in a turbulent field.

I have several fundamental issues with the manuscript that makes it not publishable in its current form. There are some small copy editing aspects that will also need to be addressed. I did not list them here since I assume the manuscript will be rewritten.

The theoretical approach is not novel. Gerritsen and Stickler (1977) modeled encounter rates of copepod based on their relative swimming speeds. Osborn and Rothshild, 1988 added turbulence to the equation. Since then (nearly 30 years ago) there have been updates to the models including the effects of turbulence on the perceptive field. Most study have assumed a parabolic shape of encounter success with increased turbulence. The initial rise in the encounter rate is due to the added relative velocity due to turbulence. The decrease in successful encounter at high turbulence is driven initially by the decrease in perception distance and then, at very high turbulence levels, by the inability of the copepods to react to each other within the short time frame of the encounter. The model shown in this study does not appear to advance our current understanding of how turbulence may enhance or diminish mating in copepods.

The authors state that " We study this important yet unexplored aspect of plankton ecology in the context of mating and show that self-locomotion creates large differences in the inward component of the pairwise radial relative velocity of copepods, which directly results in larger encounter rates at separation distances comparable to the spatial extend (this should be extent) of the flow field they generate while swimming (Kiorboe et al., 2014)"

This is simply not true. There is a rich history in the literature of looking at the effects of turbulence on the encounter rate of copepods. While not all of the work deals specifically with mating, it is described as encounter rate, or capture volume as the authors have done in this study.

In terms of the biological experiments in which copepods were observed and tracked in a turbulent regime, I have 2 suggestions that would make this manuscript stronger.

1) It would be valuable to quantify the ability of male copepods to actually grab and mate with female copepods under different levels of turbulence. Currently based on Figures 1-4 it is unclear if males and females were individually identified. This is important because females don't track males or other females. Only the males do the mate tracking. It also remains unclear if the males actually grabbed on to the females and mated or did they just get close to each other in the video? If the authors can identify the females and males in the video (using swimming patterns), the number of successful mating events could be quantified. This would be valuable data.

2) I am concerned about the level of turbulence used in this study. The copepods used in this study are on the order of 1.2 mm. The Kolmogorov length scale used in this study was 0.450 mm. The time scale was 0.2s. These turbulence levels are extremely high – not just at the high end of ocean turbulence. These levels are on the order of a tidal bore. The supplementary figure (Appendix 1—figure 3) shows a male copepod tracking a female's chemical trail. This mate tracking behavior suggests that these copepods are evolved to mate in relatively quiet waters. The length scales shown in this figure define a volume of 100 mls. At the turbulence levels used in this study, the eddy size would be 0.045^3^ cm^3^ or 0.091 nanoliters. This is a tiny volume. At this level the trails laid by the female would be destroyed. The experiments need to be done at more moderate levels of turbulence to be realistic for this species.

Finally as a closing comment – the capture volume of copepods in turbulence is linearly related to the swimming speed of the copepods and the added speed caused by turbulence (Rothschild and Osborn). However the encounter rate is exponentially related to the perceptive distance. This has a huge impact of the ability of copepods to find each other. Currently, not much is known about the effects of turbulence on the mechano-receptive or chemo-receptive abilities of copepods. However it is likely that they are impacted by turbulence. Since the encounter rate is so highly dependent on this parameter is difficult to model the outcome without a kernel that parameterizes it impact.

Gerritsen, J. and Strickler, J. R. (1977) Encounter probabilities and community structure in zooplankton: a mathematical model. J. Fish. Res. Board Can., 34, 73-82.

B.J. Rothschild, T.R. Osborn Small-scale turbulence and plankton contact rates

Journal of Plankton Research, Volume 10, Issue 3, May 1988, Pages 465-474, https://doi.org/10.1093/plankt/10.3.465

Kiørboe T, MacKenzie BR (1995) Turbulence-enhanced prey encounter rates in larval fish: effects of spatial scale, larval behaviour and size. J Plankton Res 17:2319-2331

Lewis DM (2003) Planktonic encounter rates in homogeneous isotropic turbulence: the case of predators with limited fields of sensory perception. J Theor Biol 222:73-97

Reviewer #2:

This is an interesting paper dealing with the ability of calanoid copepods to find mates in a turbulent environment. The study is based on a combination of analytical models and careful experimental studies. It is found that copepods have an enhanced encounter rate in turbulent environments compared to than in calm waters. The conclusion is consistent with a number of related investigations dealing with encounters in turbulence. The topic of the study is thus not entire new, but the methods and results are well described and add to the existing knowledge in the field. As far as I can see, the paper is giving adequate credit to previous studies. I can, in principle, recommend publication of the paper, but have some comments for the authors. Some need their attention, and some I think should be considered.

1) Figure 1 needs attention. The caption mentions some cones, that I do not find, please clarify. The figure contains "blur" in blue colour: I presume this should represent turbulence? It is not explained. In part due to this blue colour, the colour coding of the trajectories becomes unclear: I had to enlarge the figure significantly to see this. Of course any reader can do the same, but only in the electronic version. The figure is important in my opinion, so it deserves more attention. Figure 1C has interesting information, this comes out quite clearly.

2) The authors discuss finite inertia of the copepods (an entire section is devoted to this) but not much their finite size which is mentioned only in bypassing, e.g. caption of Appendix 1—figure 1. It seems to me that organisms are discussed as being point particles. The mass density of copepods is not much different from that of water, after all, but the finite size can be important and have been discussed in the literature. Finite size effects may not be dramatic for spherical particles, but elongated ones are likely to cluster in turbulent flows (this is actually consistent with results given by the black symbols for dead copepods in Figure 4). The authors refer to comparisons of measurements for dead and live copepods, but it is not clear how they make use of this information. I agree though that the finite-size information is present in this measurement.

3) I would have expected the elongation of pheromone traces due to turbulence to be important. Indeed, it is mentioned in the Introduction. Since the flow is incompressible, the volume of a trace is constant, and since it elongates (nearly exponentially, e.g. Batchelor, "The effect of homogeneous turbulence on material lines and surfaces", Proc. R. Soc. London, Ser. A 213, 349 (1952)) it must thin out quite rapidly and some places be too thin to be detectable. The pheromone traces will in reality break-up into several smaller ones. I would have expected discussions like this to take more place. The authors might consider the point?

4) As far as I can see, it is assumed that copepods will encounter as soon as they enter within a suitably defined "region of interaction", this is how I read the discussion of Figure 1B, for instance. MacKenzie et al.,. "Evidence for a dome-shaped relationship between turbulence and larval fish ingestion rates", Limnol. Oceanogr. 39, 1790 (1994) have argued that even in case two organisms come within a contact range, it might still happen that turbulent motions separate them again to reduce the probability of an ultimate encounter. It seems that this model has found some support (Pecseli et al. "Feeding of plankton in turbulent oceans and lakes", Limnol. Oceanogr. 64, 1034 (2019)). Can this scenario have some bearing for the authors problem?

5) Finally, although the disposition of the paper is good, it is at places difficult to read (at least for me). Some sentences can be very long, often over 3 or more lines of text. It might be advisable to go through the manuscript once more and break up such long sentences.

[Editors’ note: further revisions were suggested prior to acceptance, as described below.]

We have asked the opinion of a third reviewer, and the discussion reached the same conclusion as the original peer review. Specifically, all reviewers agree that both the model and the data are of great quality and advance the state of the art. However, as you also note in your appeal letter, the results demonstrate that males attempt at reaching other individuals, but there is no evidence that mating or even contact actually occurs.

The relevance of the findings for the biology of copepods hinges on the assumption that mating eventually occurs. We would be thrilled to reconsider this work as a new submission if you were to provide evidence to support this assumption.

The rest of the comments in the individual reviews were attached to help you improve the work, but were individual viewpoints of the reviewers and not shared not crucial for the decision.

Reviewer #3:

This paper reports on a study on the encounter dynamics of planktonic copepods in turbulent flows. It presents state-of-the-art measurements on the simultaneous tracking of copepods and fluid tracers in turbulence, which allow to assess the approaching velocity of copepods pairs down to a scale of 1mm (approximately one body length). The average properties of the approaching process are interpreted by means of an empirical statistical model. The conclusions drawn from the study, and in particular their biological implications on the efficiency of mate finding, are in my view not sufficiently supported by the experimental evidence, as the actual mating events are not measured experimentally.

Here below, my three main remarks on this study:

1) The paper contains new measurements and a number of interesting and solid results on the mechanistic (physical) aspects of copepod dynamics. The measurements reported in Figure 2 convey in a convincing way the message that the active motion of copepods boosts their approach.

2) One main limitation of the experimental technique is the inability to detect effective contact among copepods. This limitation makes the biological considerations about mating rather speculative.

3) The probabilistic model of plankton motion at difference from previously existing models neglects the existence of correlations between the swimming behavior (cruising, jumping) and turbulent cues (strain, shear or rotation dominated events). This is a central question to which many studies have been devoted in the past. Can this assumption be tested against the experimental data? In the preset setup the simultaneous tracking of copepods and fluid tracers could in principle allow to characterize the turbulence in the vicinity of a copepod.

In the light of the limitation at point 2), I judge the article not suitable for publication in *eLife*. However, the authors shall be definitely encouraged to publish this work in a different form.

---

## [Author Response]

All reviewers agree that both the model and the data are of great quality and advance the state of the art. However, as you also note in your appeal letter, the results demonstrate that males attempt at reaching other individuals, but there is no evidence that mating or even contact actually occurs. The relevance of the findings for the biology of copepods hinges on the assumption that mating eventually occurs. We would be thrilled to reconsider this work as a new submission if you were to provide evidence to support this assumption.

We understand that it is important to demonstrate that males actually engage in mating. We have overcome the limitation of our previous automated tracking approach by manually tracking individuals in contact with each other and we now provide the requested evidence. We have included a new figure (Figure 4 in the revised manuscript) showing a complete mating event in turbulence. The figure shows the sequence of steps leading to mating: the independent motion of the two copepods and the advection by the flow that together bring individuals within their perception radius, the detection of the female by the male, the immediate reorientation of the male, the pursuit, the capture, and finally the tumbling behavior that occurs when the male attempts at transferring its spermatophore. Together with the observation of many mating events in the original image sequences, this figure supports the assumption that copepods do mate efficiently in turbulence. We believe that this new figure completely addresses the concern of the first and third reviewer. We also wrote in the main text (in the Results and Discussion section):

"The inward motion of the male within the perception radius leads to contact between the two organisms of the pair. […] A large number of mating events similar to the one shown in Figure 4 are visible in the image sequences, indicating that mating does occur frequently in turbulence."

Reviewer #1:1) The theoretical approach is not novel. Gerritsen and Stickler (1977) modeled encounter rates of copepod based on their relative swimming speeds. Osborn and Rothshild, 1988 added turbulence to the equation. Since then (nearly 30 years ago) there have been updates to the models including the effects of turbulence on the perceptive field. Most study have assumed a parabolic shape of encounter success with increased turbulence. The initial rise in the encounter rate is due to the added relative velocity due to turbulence. The decrease in successful encounter at high turbulence is driven initially by the decrease in perception distance and then, at very high turbulence levels, by the inability of the copepods to react to each other within the short time frame of the encounter. The model shown in this study does not appear to advance our current understanding of how turbulence may enhance or diminish mating in copepods.

We are very familiar with the pioneering work of Rothschild and Osborn (1988) and with its limitations. The study of Rothschild and Osborn (1988) builds on the previous work by Gerritsen and Strickler (1977) who studied the encounter rates of species that move independently in random directions and for which the velocity magnitude follows a given distribution. Rothschild and Osborn (1988) introduced turbulent velocity difference on top of organism motion. They used squared velocity difference instead of the radial component of the difference, which is a non-rigorous approximation since particles (and plankton) move in the same coherent small-scale eddy. Therefore, the work of Rothschild and Osborn (1988) is applicable only for the species that was considered by Gerritsen and Strickler (1977) and is accurate by an order of magnitude only. Our work builds on real biological data to offer a model that is accurate up to approx. 20 % in terms of encounter rate and that can be applied to a wide range of species because it only requires knowledge of the probability density function of the magnitude of the velocity in calm water, which is straightforward to obtain empirically. Given the large offset we had opted not to include the references to Rothschild and Osborn (1988) and Gerritsen and Strickler (1977) in the submitted manuscript. We recognize now from the comments of the reviewer that this was not the best strategy and have included a note on these pioneering studies in the revised manuscript. We have also added references to earlier studies that have built on the work of Rothschild and Osborn (1988) and Gerritsen and Strickler (1977). More specifically, we wrote:

"The first model (Rothschild and Osborn 1988) derives from a formulation that considers organisms swimming independently in random direction and in calm water (Gerritsen and Strickler 1977). […] These models have allowed oceanographers to study the implications of turbulence-enhanced contact rates on grazing, predation, and optimal strategies in the plankton (MacKenzie et al., 1994; Lewis and Pedley, 2001, Lewis, 2003; Visser et al., 2009; Pécseli et al., 2019) and they often show good agreement between analytical results and field or laboratory data (Pécseli et al., 2019)."

We are also very familiar with the dome-shaped relationship between turbulence intensity and capture rates. It is often assumed that, after an initial increase, the number of successful encounters in turbulence decreases because of a decrease in perception distance and because of the inability of the organisms to react to each other within the short time frame of the encounter. This trend was first recognized in theoretical studies of fish larvae feeding on zooplankton and was later confirmed by laboratory and field data. We have added two notes on this specific point in the revised manuscript and we have also included more relevant bibliography (in the Introduction and in the Results and Discussion section). However, our results show that (a) calanoid copepods are capable to detect and pursue nearby organisms entering their reactive zone even in relatively strong turbulence and within the short time frame of the encounter, and (b) that the distance at which inward motion between nearby organisms takes place is similar in calm water and turbulence, which reveals the surprising ability of copepods to differentiate between the hydrodynamic signals generated by a nearby conspecific and the background noise generated by turbulence. These results are in sharp contrast with the comment of the reviewer. We believe that our results add substantially to our understanding or plankton encounter rates in turbulence. This information is now mentioned in the manuscript.

In addition, our model includes not only the contribution of organism motility and turbulent velocity fluctuations in encounter rates, but also the contribution of preferential concentration due to organism density, finite size, and elongated shape, via the term 𝑔(𝑟) in the geometric collision kernel 𝐾(𝑟) = 𝛾(𝑟)𝑔(𝑟). Therefore, our model is more complete than earlier formulations that have neglected the coupling between small-scale eddies and the inertia and shape of the organisms. Our model is directly applicable to species for which inertia and shape result in clustering, allowing accurate predictions when studying predation, interactions, and resource exploitation in the plankton. This information is given in the revised manuscript.

2) The authors state that "We study this important yet unexplored aspect of plankton ecology in the context of mating and show that self-locomotion creates large differences in the inward component of the pairwise radial relative velocity of copepods, which directly results in larger encounter rates at separation distances comparable to the spatial extend (this should be extent) of the flow field they generate while swimming (Kiorboe et al., 2014)." This is simply not true. There is a rich history in the literature of looking at the effects of turbulence on the encounter rate of copepods. While not all of the work deals specifically with mating, it is described as encounter rate, or capture volume as the authors have done in this study.

We entirely agree with the reviewer that there is a rich literature on the effects of turbulence on the encounter rates of plankton. It was not our intention to claim priority on this issue and we have modified the text accordingly. However, to our knowledge, most of the work relevant to our study (encounter rates between organisms of similar swimming capabilities, therefore excluding studies on copepods feeding on microalgae) are theoretical or numerical, and none have addressed the specific case of mating. We have nevertheless added references to earlier studies in the revised version of the manuscript (in the Introduction and in the Results and Discussion section).

3) It would be valuable to quantify the ability of male copepods to actually grab and mate with female copepods under different levels of turbulence. Currently based on Figures 1-4 it is unclear if males and females were individually identified. This is important because females don't track males or other females. Only the males do the mate tracking. It also remains unclear if the males actually grabbed on to the females and mated or did they just get close to each other in the video? If the authors can identify the females and males in the video (using swimming patterns), the number of successful mating events could be quantified. This would be valuable data.

We understand that it is important to demonstrate that males actually engage in mating. In the first version of the manuscript, while we demonstrated that males attempted at reaching nearby individuals within the perception distance, there was no evidence that mating actually occurred. We have overcome the limitation of our previous automated tracking approach by manually tracking individuals in contact with each other. We now provide evidence that inward motion leads to contacts and that contacts lead to mating. We have prepared a new figure (Figure 4 in the revised manuscript) showing a complete mating event in turbulence. The figure shows the sequence of steps leading to mating: the independent motion of the two copepods and the advection by the flow that together bring individuals within their perception radius, the detection of the female by the male, the immediate reorientation of the male, the pursuit, the capture, and finally the tumbling behavior that occurs when the male attempts at transferring its spermatophore. Together with the observation of many mating events in the original image sequences, this figure supports the assumption that copepods do mate efficiently in turbulence. We also wrote in the main text (in the Results and Discussion section):

"The inward motion of the male within the perception radius leads to contact between the two organisms of the pair. […] A large number of mating events similar to the one shown in Figure 4 are visible in the image sequences, indicating that mating does occur frequently in turbulence."

We have also clarified in the revised manuscript (in the Results and Discussion section) that the minimal separation distance at which two organisms could be detected via our automatic particle tracking technique (1 mm) corresponds to the distance between the centroids of the two organisms. Because copepods are approximately 1 mm in size, this means that we can actually resolve the approach up to contact for thousands of events.

Our work quantifies the contribution of organism motility and physical effects due to fluid flow (turbulent advection and clustering due to density, finite size, and elongated shape) on the encounter rate of copepods swimming in turbulence. We document both the flux of organisms around swimming copepods for different perception distances and the behavioral response that correlates the motion of two organisms once motility and turbulence bring them in close proximity. We were able to resolve interactions between organisms up to contact events, for thousands of events, and we also show in the new Figure 4 that inward motion leads to contacts and that organisms engage in actual mating. Our manuscript now combines results obtained via an automatic particle tracking technique that allows us to obtain previously unavailable data at very high degree of statistical reliability and up to contact between organisms, with qualitative data that proves that contact leads to mating.

4) I am concerned about the level of turbulence used in this study. The copepods used in this study are on the order of 1.2 mm. The Kolmogorov length scale used in this study was 0.450 mm. The time scale was 0.2s. These turbulence levels are extremely high – not just at the high end of ocean turbulence. These levels are on the order of a tidal bore. The supplementary figure (Appendix 1—figure 3) shows a male copepod tracking a female's chemical trail. This mate tracking behavior suggests that these copepods are evolved to mate in relatively quiet waters. The length scales shown in this figure define a volume of 100 mL. At the turbulence levels used in this study, the eddy size would be 0.045^3^ cm^3^ or 0.091 nanoliters. This is a tiny volume. At this level the trails laid by the female would be destroyed. The experiments need to be done at more moderate levels of turbulence to be realistic for this species.

A large body of literature indicates that the level of turbulence used in our measurements is representative of the hydrodynamic conditions experienced by *Eurytemora affinis* and many other species of copepods in their estuarine and coastal habitats where most of the turbulence comes from tidal forcing (Schmitt et al., 2011; MacMillan et al., 2016; Tu et al., 2019; Ross et al., 2019). The turbulence level is also well within the range considered in previous laboratory and theoretical studies of plankton motion in turbulence (e.g., Yen et al., 2008). This information is now indicated in the revised manuscript, together with references to field and laboratory studies showing that our intensity of turbulence is very realistic for the species used in our work and similar to that used in earlier measurements.

McMillan J. M., Hay A. E., Lueck R. G., Wolk F. (2016) Rates of dissipation of turbulent kinetic energy in a high Reynolds number tidal channel. Journal of Atmospheric and Oceanic Technology, 33:817-837.

Ross L., Huguenard K., Sottolichio A. (2019) Intratidal and fortnightly variability of vertical mixing in a macrotidal estuary: the Gironde. JGR Oceans, 124(4):2641-2659.

Schmitt F. G., Devreker D., Dur G., Souissi S. (2011) Direct evidence of tidally oriented behavior of the copepod *Eurytemora affinis* in the Seine estuary. Ecological Research, 26:773-780.

Tu J., Fan D., Zhang Y., Voulgaris G. (2019) Turbulence, sediment‐induced stratification, and mixing under macrotidal estuarine conditions (Qiantang estuary, China). JGR Oceans, 124(6):4058-4077.

Yen J., Rasberry K. D., Webster D. R. (2008) Quantifying copepod kinematics in a laboratory turbulence apparatus. Journal of Marine Systems, 69:283-294.

Conducting measurements in calmer hydrodynamic conditions so that pheromone trails are not destroyed, as suggested by the reviewer, would not advance our understanding of the behavioral ecology of plankton. Previous empirical studies on copepod mating have already considered encounters in calm water. The case of mating in turbulence was never investigated, primarily because of the difficulties of resolving the motion of plankton swimming in three dimensions, in turbulence, and at appropriate spatial and temporal scales. The hydrodynamic conditions in these studies are very different from those experienced by copepods in their environment, where turbulence can very over six orders of magnitude (Yamazaki and Squires, 1996). Consequently, most of what we know about the ability of copepods to find mates and reproduce derives from observations conducted in flow conditions that are not always environmentally relevant. Empirical evidence for mate-tracking behavior in calm hydrodynamic conditions has shaped the present understanding of plankton ecology for decades (e.g., the seminal work of Yen et al., 1998 and Bagøien and Kiørboe, 2005) and has led some oceanographers to assume that copepods are evolved to mate in relatively quiet waters and are unable to mate in turbulence, although this was never confirmed.

Our empirical results show, for the first time, that this vision is largely incomplete. They suggest that reproduction in copepods is not restricted to spatial and temporal windows of low turbulence intensity, which differs from what the community has assumed over the past two decades. The ability of copepods to find mates efficiently in turbulence relaxes the boundaries between quiescent and turbulent conditions in terms of individual fitness, and suggests that copepods are well adapted to the intermittent hydrodynamic conditions that they experience in their dynamic environment. This finding has large implications for our understanding of copepod ecology and of the relationships between turbulence intensity in the environment and the fitness of copepod populations. This information is given in the revised manuscript.

Therefore, we do not agree with the reviewer that measurements should be conducted in calmer conditions. Such measurements would repeat previous studies, including our own (e.g., Yen et al., 1998; Bagøien and Kiørboe, 2005; Dur et al., 2011). They would provide information that is already well known, they would not allow studying how copepods find mates in flow conditions that are more environmentally relevant, and consequently they would not provide the new perspective that this work offers. We have made all possible efforts to evidence that mating occurs in turbulence. First, we have analyzed in depth the behavioral and physical mechanisms that lead to encounters. By measuring simultaneously the motion of copepods and flow tracers, we have quantified the contributions of motility, preferential concentration, and flow advection in driving enhanced encounter rates in turbulence. We document at unprecedented accuracy (tens of millions of data points) both the flux of organisms around swimming copepods and the behavioral response that correlates the motion of two organisms once motility and turbulence bring them in close proximity. Secondly, we were able to resolve interactions between organisms up to contact, for thousands of events, and we show in additional measurements using separate genders that the inward motion within the radius of perception is meditated by males only. Thirdly, we show that contacts lead to actual mating. Parametric studies over a range of turbulence intensities are beyond the scope and can be addressed in future work.

Bagøien E., Kiørboe T. (2005) Blind dating – mate finding in planktonic copepods. I. Tracking the pheromone trail of *Centropages typicus*. Marine Ecology Progress Series, 300:105-115.

Dur G., Souissi S., Schmitt F. G., Beyrend-Dur D., Hwang J.S. (2011) Mating and mate choice in *Pseudodiaptomus annandalei* (Copepoda: Calanoida). Journal of Experimental Marine Biology and Ecology, 402:1-11.

Yamazaki H., Squires KD. (1996) Comparison of oceanic turbulence and copepod swimming. Marine Ecology Progress Series, 144:299.

Yen J., Weissburg M. J. Doall M. H. (1998) The fluid physics of signal perception by mate-tracking copepods. Philosophical Transactions of the Royal Society B, 353(1369):787:804.

5) Finally as a closing comment – the capture volume of copepods in turbulence is linearly related to the swimming speed of the copepods and the added speed caused by turbulence (Rothschild and Osborn). However the encounter rate is exponentially related to the perceptive distance. This has a huge impact of the ability of copepods to find each other. Currently, not much is known about the effects of turbulence on the mechano-receptive or chemoreceptive abilities of copepods. However it is likely that they are impacted by turbulence. Since the encounter rate is so highly dependent on this parameter is difficult to model the outcome without a kernel that parameterizes it impact.

The reviewer claims that turbulence is likely to impact the perception ability of male copepods, and therefore that it is difficult to model encounter rates without a kernel that parametrizes its impact. In this study, we provide experimental evidence that the perception distance of copepods is similar (approx. 4 mm) in calm water and in turbulence. This point is clearly indicated in the text where we wrote:

"Surprisingly, the distance 𝑟 ≈ 4 mm (corresponding to approximately 8 𝜂) at which we observe directed motion between organisms is similar in calm water and in turbulence, which reveals the intriguing ability of copepods to correctly identify the hydrodynamic signals generated by a conspecific among the background noise caused by turbulence. […] Our results show that, in this region, an organism appears to be able to distinguish between the flow disturbances created by another organism as it swims and the linear velocity gradients of turbulence within 𝜂_0_."

We feel sorry that the reviewer seems to have missed this information. Besides, extension of our model to include perception is straightforward because we can again follow the analogy with rain formation and have an efficiency factor that multiplies the collision kernel. This point is indicated in the manuscript where we wrote:

"The second term of the collision kernel, termed the collision efficiency, provides corrections to 𝐾(𝑟) that account for interactions between organisms at 𝑟 below the range of validity of 𝐾(𝑟), that is, for 𝑟 ≤ 4 mm within the clearance volume. […] Therefore, the perception capability appears to be relatively unaffected for this species, and we can already anticipate that the encounter rate will not vary substantially."

Therefore, we already give the framework for the extension of the model in this paper. If the perception ability is unaffected (as suggested by males moving inward, starting at the same separation distance in calm water and in turbulence), we can anticipate that the encounter rate remains unchanged, again in opposition to what the reviewer claims.

Reviewer #2:This is an interesting paper dealing with the ability of calanoid copepods to find mates in a turbulent environment. The study is based on a combination of analytical models and careful experimental studies. It is found that copepods have an enhanced encounter rate in turbulent environments compared to than in calm waters. The conclusion is consistent with a number of related investigations dealing with encounters in turbulence. The topic of the study is thus not entire new, but the methods and results are well described and add to the existing knowledge in the field. As far as I can see, the paper is giving adequate credit to previous studies. I can, in principle, recommend publication of the paper, but have some comments for the authors. Some need their attention, and some I think should be considered.

We thank the reviewer for their constructive comments and are happy to hear that they find the paper interesting. While it is true that plankton encounter rates in turbulence have already been studied, to our knowledge most of the earlier work is theoretical or numerical (e.g., Pécseli et al., 2014). In this study, we take advantage of an advanced particle tracking technique to obtain direct experimental results at unprecedented accuracy. We prove that two key conditions for copepod mating in turbulence are met. Firstly, the ability to reach the radius of perception at higher rates than in still conditions thanks to the contribution of active motion and turbulence advection. Secondly, the ability to react to each other via directed motion within the perception radius and within the short time frame of the encounter. We then show that copepods convert high encounter rates to frequent contact events and then to mating (new Figure 4 in the revised manuscript), and therefore that they are able to mate even when turbulence is strong. The first point is therefore in good agreement with the literature, but the second and third bring new perspectives to the issue. Indeed, our observations call for a reappraisal of our current knowledge of the behavioral ecology of calanoid copepods. Empirical evidence for mate-tracking behavior in calm hydrodynamic conditions has shaped the present understanding of copepod behavioral ecology for decades (e.g., the seminal work of Yen et al., 1998 and Bagøien and Kiørboe, 2005) and it is often assumed that copepods are evolved to mate in relatively quiet waters, although this was never confirmed. Our empirical results show that this vision is largely incomplete. We provide responses to specific comments below.

Bagøien E., Kiørboe T. (2005) Blind dating – mate finding in planktonic copepods. I. Tracking the pheromone trail of *Centropages typicus*. Marine Ecology Progress Series, 300:105-115.

Pécseli H. L., Trulsen J. K., Fiksen Ø. (2014) Predator–prey encounter and capture rates in turbulent environments. Limnology and Oceanography: Fluids and Environment, 4(1):85-105.

Yen J., Weissburg M. J. Doall M. H. (1998) The fluid physics of signal perception by mate-tracking copepods. Philosophical Transactions of the Royal Society B, 353(1369):787:804.

1) Figure 1 needs attention. The caption mentions some cones, that I do not find, please clarify. The figure contains "blur" in blue colour: I presume this should represent turbulence? It is not explained. In part due to this blue colour, the colour coding of the trajectories becomes unclear: I had to enlarge the figure significantly to see this. Of course any reader can do the same, but only in the electronic version. The figure is important in my opinion, so it deserves more attention. Figure 1C has interesting information, this comes out quite clearly.

The blue cones indeed represent the direction and magnitude of the flow velocity vectors. Their length and opacity are determined by the trade-off between achieving clearly identifiable individual cones and being able to see the copepod trajectories. Making the cones larger would results in poorly visible copepod trajectories, and making them smaller would blur the flow field and make this information lost to the reader. We have tried to reach an appropriate trade-off in Figure 1 and in the new Figure 4, and we wrote in the caption of Figure 1:

"Because the cones are semi-transparent to allow visualizing the copepod trajectories, it may be necessary to zoom in to see them individually."

2) The authors discuss finite inertia of the copepods (an entire section is devoted to this) but not much their finite size which is mentioned only in bypassing, e.g. caption of Appendix 1—figure 1. It seems to me that organisms are discussed as being point particles. The mass density of copepods is not much different from that of water, after all, but the finite size can be important and have been discussed in the literature. Finite size effects may not be dramatic for spherical particles, but elongated ones are likely to cluster in turbulent flows (this is actually consistent with results given by the black symbols for dead copepods in Figure 4). The authors refer to comparisons of measurements for dead and live copepods, but it is not clear how they make use of this information. I agree though that the finite-size information is present in this measurement.

We thank the reviewer for drawing our attention to this point. We have mentioned in the revised manuscript that shape, in addition to size and density, may also influence the dynamics of copepods in turbulence and their preferential concentration. We show in Figure 5 that the clustering of dead copepods due to finite size, density slightly larger than seawater, and elongated shape is negligible and not likely to increase the flux of organisms within their perception radius. This information is given in the revised manuscript. We stress that the clustering analysis shown in Figure 5 quantifies the combined effects of density, finite size, and elongated shape, as mentioned by the reviewer.

We have decided to show the result of the clustering analysis for living copepods to maintain consistency throughout the Results and Discussion section. Indeed, all measurements have been conducted on tracers, inert carcasses, and living copepods both in calm water and in turbulence. We believe that the result of the clustering analysis for living copepods in turbulence is important because it shows clearly the signature of individuals moving toward each other at small separation distance, in good agreement with the analysis of the pairwise radial relative velocity. This information is given in the text.

3) I would have expected the elongation of pheromone traces due to turbulence to be important. Indeed, it is mentioned in the Introduction. Since the flow is incompressible, the volume of a trace is constant, and since it elongates (nearly exponentially, e.g. Batchelor, "The effect of homogeneous turbulence on material lines and surfaces", Proc. R. Soc. London, Ser. A 213, 349 (1952)) it must thin out quite rapidly and some places be too thin to be detectable. The pheromone traces will in reality break-up into several smaller ones. I would have expected discussions like this to take more place. The authors might consider the point?

We thank the reviewer for their suggestion. We have added a note on this specific point where we wrote:

"However, under conditions involving turbulence, pheromone trails are bound to elongate exponentially because of the incompressibility of the flow (Batchelor, 1952), to thin out rapidly and become too thin to be detectable, and then to break within seconds or less into disconnected filaments of concentrated cues separated by gaps with no detectable signal until further mixing and molecular diffusion cause local variations of the odorant to vanish (Shraiman and Siggia, 2000)."

4) As far as I can see, it is assumed that copepods will encounter as soon as they enter within a suitably defined "region of interaction", this is how I read the discussion of Figure 1B, for instance. MacKenzie et al., "Evidence for a dome-shaped relationship between turbulence and larval fish ingestion rates", Limnol. Oceanogr. 39, 1790 (1994) have argued that even in case two organisms come within a contact range, it might still happen that turbulent motions separate them again to reduce the probability of an ultimate encounter. It seems that this model has found some support (Pecseli et al. "Feeding of plankton in turbulent oceans and lakes", Limnol. Oceanogr. 64, 1034 (2019)). Can this scenario have some bearing for the authors problem?

We thank the reviewer for drawing our attention to this important point. We are familiar with the dome-shaped relationship between turbulence intensity and capture rates. It is often assumed that, after an initial increase, the number of successful encounters in turbulence decreases because of a decrease in perception distance and because of the inability of the organisms to react to each other within the short time frame of the encounter. This trend was first recognized in theoretical studies of fish larvae feeding on zooplankton and was later confirmed by laboratory and field data, for instance in the study cited by the reviewer. We have added two notes on this specific point in the revised manuscript and we have also included the relevant bibliography (in the Introduction and in the Results and Discussion section). However, our results show that (a) calanoid copepods are capable to detect and pursue nearby organisms entering their reactive zone even in strong turbulence and within the short time frame of the encounter, and (b) that the distance at which inward motion takes place is similar in similar in calm water and turbulence, which reveals the surprising ability of copepods to differentiate between the hydrodynamic signals generated by a nearby conspecific and the background noise generated by turbulence. We believe that these two important results add substantially to our understanding or plankton encounter rates in turbulence. This information is now mentioned in the manuscript.

5) Finally, although the disposition of the paper is good, it is at places difficult to read (at least for me). Some sentences can be very long, often over 3 or more lines of text. It might be advisable to go through the manuscript once more and break up such long sentences.

We have taken this concern very seriously and have broken long sentences that were difficult to read into several smaller ones, throughout the manuscript.

Reviewer #3:1) The paper contains new measurements and a number of interesting and solid results on the mechanistic (physical) aspects of copepod dynamics. The measurements reported in Figure 2 convey in a convincing way the message that the active motion of copepods boosts their approach.

We thank the reviewer for their comments and are excited to hear that they found our results interesting and solid.

2) One main limitation of the experimental technique is the inability to detect effective contact among copepods. This limitation makes the biological considerations about mating rather speculative.

As indicated above in our response to the Editor and to the comments of reviewer 1, we understand that it is important to demonstrate that males actually engage in mating. In the first version of the manuscript, while we demonstrated that males attempted at reaching nearby individuals within the perception distance, there was no evidence that mating actually occurred. We have overcome the limitation of our previous automated tracking approach by manually tracking individuals in contact with each other. We now provide evidence that inward motion leads to contacts and that contacts lead to mating. We have prepared a new figure (Figure 4) showing a complete mating event in turbulence. The figure shows the sequence of steps leading to mating: the independent motion of the two copepods and the advection by the flow that together bring individuals within their perception radius, the detection of the female by the male, the immediate reorientation of the male, the pursuit, the capture, and finally the tumbling behavior that occurs when the male attempts at transferring its spermatophore. Together with the observation of many mating events in the original image sequences, this figure supports the assumption that copepods do mate efficiently in turbulence. We also wrote in the main text (in the Results and Discussion section):

"The inward motion of the male within the perception radius leads to contact between the two organisms of the pair. […] A large number of mating events similar to the one shown in Figure 4 are visible in the image sequences, indicating that mating does occur frequently in turbulence."

Our manuscript now combines results obtained via an automatic particle tracking technique that allows us to obtain previously unavailable data at very high degree of statistical reliability and up to contact between organisms, with qualitative data that proves that contact leads to mating. We have also clarified in the manuscript (in the Results and Discussion section) that the minimal separation distance at which two organisms could be detected via our automatic particle tracking technique (1 mm) corresponds to the distance between the centroids of the two organisms. Because copepods are approximately 1 mm in size, this means that we can actually resolve the approach up to contact for thousands of events.

3) The probabilistic model of plankton motion at difference from previously existing models neglects the existence of correlations between the swimming behavior (cruising, jumping) and turbulent cues (strain, shear or rotation dominated events). This is a central question to which many studies have been devoted in the past. Can this assumption be tested against the experimental data? In the preset setup the simultaneous tracking of copepods and fluid tracers could in principle allow to characterize the turbulence in the vicinity of a copepod.

We are very familiar with earlier studies that have studied correlations between specific flow signals and the behavior of copepods, notably their jumps. These studies show that, in calm water or in laminar flow, copepods react to localized flow velocity gradients in their vicinity (for instance, those created by the tip of a pipette) via powerful escape jumps. These jumps are often directed away from the source of the disturbance and occur even when the disturbance is very weak (e.g., Kiørboe et al., 1997; Buskey et al., 2002; Yen et al., 2015; Buskey et al., 2017). In Michalec et al., 2017, we have already tested whether this response also occurs in turbulence by testing for correlations between the swimming behavior of copepods and turbulent cues. Using the setup described in this manuscript, we have tracked copepods swimming freely in quasi-homogeneous, isotropic turbulence, together with inert flow tracers. We have quantified turbulence quantities at each time instant along copepod trajectories. We found no one-to-one correlation between the signals generated by turbulence near the copepods (shear stress, normal stress, inertial drag stress, and vorticity) and their jump behavior or even the direction of their motion. The absence of correlation is now mentioned in the revised manuscript. This result shows that copepods exposed to realistic turbulent conditions do not react to the turbulence properties of the background flow the same as they react to localized hydrodynamic disturbances of comparable intensities in a still or laminar environment. Local events of strong turbulence do not trigger escape jumps, even when turbulence quantities are well above reported threshold values in calm water. Therefore, our model does not include correlations with the instantaneous flow signals encountered by copepods along their trajectories.

However, we also found that copepods adapt their swimming activity to the turbulence intensity by performing more frequent relocation jumps of limited amplitude that seem unrelated to localized hydrodynamic signals (Michalec et al., 2017). In other words, they increase their jump frequency as the turbulence intensity increases. The existence of frequent relocation jumps in a motion that is otherwise dominated by turbulent transport allows for the possibility of active locomotion. It enables copepods to transition from being passively advected by the flow to being capable of maintaining a certain control over their displacements despite the physical constraints that turbulence imposes on their motion. Our model does account for this behavioral mechanism via the fraction of time that a copepod is jumping (Equation 4). This fraction influences the probability density function of the relative velocity of copepods with respect to turbulence and thus their pairwise radial relative velocity (Equation 7). Therefore, our model already includes the correlation between the turbulence intensity and the behavior of copepods. This information is given in the revised manuscript.

Buskey E. J., Lenz P. H., Hartline D. K. (2002) Escape behavior of planktonic copepods in response to hydrodynamic disturbances: High speed video analysis. Mar. Ecol. Prog. Ser. 235:135146.

Buskey E. J., Strickler J. R., Bradley C. J., Hartline D. K., Lenz P. H. (2017) Escapes in copepods: Comparison between myelinate and amyelinate species. J. Exp. Biol. 220:754-758.

Kiørboe T., Saiz E., Visser A. (1999) Hydrodynamic signal perception in the copepod *Acartia tonsa*. Mar. Ecol. Prog. Ser. 179:97-111.

Michalec F.-G., Fouxon I., Souissi S., Holzner Markus (2017) Zooplankton can actively adjust their motility to turbulent flow. PNAS 52:E11199-E11207.

Yen J., Murphy D. W., Fan L., Webster D. R. (2015) Sensory-motor systems of copepods involved in their escape from suction feeding. Integr. Comp. Biol. 55:121-133.